# A replication competent *Plasmodium falciparum* parasite completely attenuated by dual gene deletion

Debashree Goswami [1,5], Hardik Patel [1,5], William Betz [1], Janna Armstrong[1], Nelly Camargo[1], Asha Patil[2], Sumana Chakravarty[2], Sean C Murphy[3], B Kim Lee Sim [2], Ashley M Vaughan [1,4], Stephen L Hoffman[2] & Stefan HI Kappe [1,4]✉

## Abstract

Vaccination with infectious *Plasmodium falciparum* (Pf) sporozoites (SPZ) administered with antimalarial drugs (PfSPZ-CVac), confers superior sterilizing protection against infection when compared to vaccination with replication-deficient, radiation-attenuated PfSPZ. However, the requirement for drug administration constitutes a major limitation for PfSPZ-CVac. To obviate this limitation, we generated late liver stage-arresting replication competent (LARC) parasites by deletion of the *Mei2* and *LINUP* genes (*mei2⁻/linup⁻* or LARC2). We show that *Plasmodium yoelii* (Py) LARC2 sporozoites did not cause breakthrough blood stage infections and engendered durable sterilizing immunity against various infectious sporozoite challenges in diverse strains of mice. We next genetically engineered a PfLARC2 parasite strain that was devoid of extraneous DNA and produced cryopreserved PfSPZ-LARC2. PfSPZ-LARC2 liver stages replicated robustly in liver-humanized mice but displayed severe defects in late liver stage differentiation and did not form liver stage merozoites. This resulted in complete abrogation of parasite transition to viable blood stage infection. Therefore, PfSPZ-LARC2 is the next-generation vaccine strain expected to unite the safety profile of radiation-attenuated PfSPZ with the superior protective efficacy of PfSPZ-CVac.

**Keywords** Malaria; Pre-erythrocytic; Genetically Attenuated Parasite Vaccine; Sporozoite; Liver Stage
**Subject Category** Microbiology, Virology & Host Pathogen Interaction

See also: D Moita & M Prudêncio

## Introduction

Malaria remains a significant global health threat, with an estimated 249 million cases and 608,000 deaths attributed to the disease in 2022 (World Health Organization, 2023a). The African region, especially countries in sub-Saharan Africa bear a disproportionate burden of malaria, accounting for 95% of global malaria cases (World Health Organization, 2023a). *Plasmodium* spp., obligate intracellular Apicomplexan parasites, are the causative organisms of malaria. Among the human malaria parasite species, *Plasmodium falciparum* (Pf) is the most deadly, responsible for the majority of malaria-related mortality, especially among pregnant women and children under the age of five (World Health Organization, 2023a).

The development of a highly efficacious malaria vaccine has the potential to significantly reduce the burden of malaria and support eradication efforts. The only WHO-recommended malaria vaccines are subunit vaccines, RTS,S/AS01 (RTS,S) (Stoute et al, 1998; Agnandji et al, 2014; Alonso et al, 2004; Kester et al, 2001; Alonso et al, 2005; Kester et al, 2009) and the related R21 (Datoo et al, 2021, 2022; World Health Organization, 2023b), which elicit antibodies that target the Pf circumsporozoite protein (PfCSP) expressed on the surface of Pf sporozoites. After over 30 years of development, RTS,S has recently been licensed for use in African children (Ejigiri and Sinnis, 2009) and has been shown to confer partial protection against clinical malaria, reducing malaria hospitalizations by 17% and severe disease by 22% in African infants (RTS, 2015). However, immunity provided by RTS,S is short-lived, wanes over time (Alonso et al, 2004, 2005; Agnandji et al, 2014), and there is currently insufficient evidence to demonstrate significant protection against Pf infection, thus limiting its potential for malaria elimination efforts. R21 performed well in reducing clinical malaria, but its efficacy against parasite infection remains unknown (Datoo et al, 2022, 2021). WHO stated there was no evidence yet that RTS,S and R21 differed in vaccine efficacy.

Attenuated whole pre-erythrocytic parasite vaccines administered as live sporozoite stages are another type of malaria vaccine (Vaughan and Kappe, 2017a). Unlike subunit vaccines, live sporozoite vaccines constitute an antigen agnostic approach as they express thousands of distinct and potentially protective antigens of mostly unknown nature (Lindner et al, 2019; Zanghi et al, 2023). Whole sporozoite based vaccines not only elicit neutralizing antibodies against sporozoite stages, which block their entry into the liver but also elicit robust CD8⁺ T cell immune responses that eliminate liver stages (LS) within hepatocytes (Hoffman et al, 2002; Roestenberg et al, 2009, 2011; Epstein et al,

[1]Center for Global Infectious Disease Research, Seattle Children's Research Institute, 307 Westlake Avenue North, Suite 500, Seattle, WA 98109, USA. [2]Sanaria Inc., 9800 Medical Center Dr., Rockville, MD 20850, USA. [3]Department of Laboratory Medicine and Pathology, University of Washington, Seattle, WA, USA. [4]Department of Pediatrics, University of Washington, Seattle, WA, USA. [5]These authors contributed equally: Debashree Goswami, Hardik Patel. ✉E-mail: stefan.kappe@seattlechildrens.org

2011; Bijker et al, 2013; Zaidi et al, 2017). Indeed, multiple studies from rodent and non-human primate models indicate that CD8[+] T cells (Schofield et al, 1987; Ewer et al, 2013; Cockburn et al, 2013, 2014; Huang et al, 2015; Olsen et al, 2018), specifically tissue resident memory T cells are critical for whole sporozoite vaccine-mediated protection (Wahl et al, 2022; McNamara et al, 2017).

One of the most extensively studied whole parasite vaccine platforms consists of radiation attenuated Pf sporozoites (SPZ), which were demonstrated in the 1970s to induce sterilizing immunity against controlled human malaria infection (CHMI) (Hoffman et al, 2002). An injectable formulation has been developed termed Sanaria® PfSPZ Vaccine, which is comprised of attenuated, metabolically active, aseptic, vialed, cryopreserved PfSPZ (Hoffman et al, 2010). PfSPZ Vaccine confers robust and durable sterilizing immunity against both homologous and heterologous CHMI in malaria naive adults (Hoffman et al, 2002; Roestenberg et al, 2009, 2011; Hoffman et al, 2010; Seder et al, 2013; Lyke et al, 2017) and against natural infection in malaria-exposed adults living in regions with intense malaria transmission (Sissoko et al, 2022; Sirima et al, 2022; Sissoko et al, 2017). PfSPZ Vaccine is the prototype of early liver stage-arresting replication-deficient (EARD) whole parasite vaccines; PfSPZ invade hepatocytes but arrest early in liver stage development before significant initiation of genome replication and differentiation (LS schizogony) due to radiation-induced DNA damage (Goswami et al, 2019).

An alternative attenuation method for live *Plasmodium* vaccines is the precise genetic modification of the parasite by deletion of genes that are critical for LS development (Vaughan and Kappe, 2017a; Goswami et al, 2019). These genetically attenuated parasites (GAP) are intrinsically and consistently attenuated, and their genetic identity is known. Numerous EARD GAPs were generated in the rodent malaria models *P. yoelii* (Py) and *P. berghei*, which provided animal model evidence that GAPs are capable of eliciting sterilizing immunity against malaria infection (De Koning-Ward et al, 1998; Mueller et al, 2005; van Dijk et al, 2005; Aly et al, 2008; VanBuskirk et al, 2009). However, owing to the significant divergence of rodent and human malaria species gene function, only a few viable Pf EARD GAPs were successfully generated in the PfNF54 strain, by targeting genes for which function is conserved between rodent and human malaria parasite species. One such Pf EARD GAP, termed Pf GAP3KO (Pf *p52⁻/p36⁻/sap1⁻*) (Mikolajczak et al, 2014), was shown to be safe and completely attenuated in a first-in-human clinical trial in which immunizations were administered by infected mosquito bite (Kublin et al, 2017). In a subsequent three or five dose immunization trial by infected mosquito bites, Pf GAP3KO vaccination conferred 50% sterile protection against homologous PfNF54 strain controlled human malaria infection (CHMI) (Murphy et al, 2022). A second Pf EARD GAP, Pf *b9⁻/sap1⁻* was syringe-administered as aseptic, purified cryopreserved Sanaria® PfSPZ-GA1 Vaccine (Roestenberg et al, 2020). PfSPZ-GA1 Vaccine was safe and well tolerated but showed much lower vaccine efficacy (12%) against CHMI than expected. These data were difficult to interpret, however, since the PfSPZ Vaccine reference arm in this trial also showed low efficacy, much less than previously reported (Epstein et al, 2017).

A transformative step in the development of GAP vaccines was the discovery of late liver stage- arresting replication competent (LARC) strains. LARC GAPs undergo extensive LS DNA and organellar replication and cell mass expansion, but developmentally arrest before differentiation into mature LS merozoites (Goswami et al, 2019, 2020). In rodent malaria models, LARC GAPs confer far superior pre-erythrocytic stage protection when compared to irradiated sporozoites or EARD GAPs (Butler et al, 2011). In addition, LARC GAPs confer protection against blood stage challenge, likely due to the expression of antigens that are shared between late LS and blood stages (Kreutzfeld et al, 2017; Vaughan and Kappe, 2017b, 2017a; Goswami et al, 2019). Further evidence supporting superior efficacy of replication competent parasite vaccination is provided by clinical trials using non-attenuated PfSPZ vaccination, administered during prophylaxis with anti-malarial drugs such as chloroquine, PfSPZ-CVac (CQ) or pyrimethamine, PfSPZ-CVac (PYR) (Mordmüller et al, 2017; Sulyok et al, 2021; Mwakingwe-Omari et al, 2021). PfSPZ-CVac (CQ) required only 12% of the sporozoite vaccination dose used for PfSPZ Vaccine to confer similar protective efficacy (Sulyok et al, 2021; Mordmüller et al, 2022) and conferred broad and durable sterilizing immunity against homologous and heterologous parasite CHMI (Mwakingwe-Omari et al, 2021). However, vaccination with fully infectious PfSPZ necessitates antimalarial drug cover to prevent causing malaria by vaccination and this is a major liability. Drugs may be less effective depending on dosing and human metabolism, vaccinated individuals may not comply with medication, and in case of PfSPZ-CVac (CQ), individuals suffer transient symptoms of malaria after the first dose (Sulyok et al, 2021; Murphy et al, 2021). To obviate these liabilities but retain replication competence of the whole parasite immunogen, we pursued the generation of a PfLARC GAP.

We recently reported that deletion of a *Plasmodium* gene encoding a RNA-binding protein called Mei2 (Pf *mei2⁻*), resulted in severe late LS attenuation in the FRG huHep humanized mouse model (Goswami et al, 2020). However, occasional blood stage breakthrough infections were previously observed in the Py rodent malaria model when highly susceptible BALB/cByJ mice were given a high dose of Py *mei2⁻* sporozoites (Vaughan et al, 2018). The observations of breakthrough infections in single gene deletion GAPs spurred the identification of additional gene deletions suitable for late LS attenuation and the pursuit of combination gene deletions to achieve full attenuation (Goswami et al, 2022a). In this study, we report the generation and biological characterization of PfLARC2 and PyLARC2, which carry dual gene deletions of the *PlasMei2* and *LINUP* genes. We observe that LARC2 behaves as a synthetic lethal that undergoes late LS developmental arrest and shows no evidence of causing viable breakthrough blood stage infection for both PfLARC2 in humanized mice and in the PyLARC2 in rodent malaria model. PfSPZ-LARC2 injectable vaccine formulations have been generated and the data presented herein provide the preclinical basis for clinical development of a PfSPZ-LARC2 vaccine.

## Results

### *P. yoelii* LARC2 (*mei2⁻/linup⁻*) shows complete late liver stage attenuation

*Plasmodium* (Plas) Mei2 is orthologous to the fission yeast, *Schizosaccharomyces pombe* RNA-binding protein Mei2 (Meiosis inhibited-2). PlasMei2 expression is LS-specific, and it exhibits a granular cytoplasmic localization pattern in mid-to-late LS

schizonts (Dankwa et al, 2016; Goswami et al, 2020). We have previously reported that PlasMei2 is essential for late LS differentiation. Deletion of *PlasMei2* in Py (Dankwa et al, 2016) and Pf (Goswami et al, 2020), resulted in severe attenuation of late LS development and a lack of LS merozoite formation in both parasite species. However, infection of highly susceptible BALB/cByJ mice with doses of 200,000–500,000 Py *mei2⁻* sporozoites resulted in blood stage breakthroughs in approximately 10% of the mice, indicating severe but incomplete attenuation (Vaughan et al, 2018). In search of additional loci that are important for late LS development, we recently identified a LS-specific protein that localizes to the liver stage nucleus named liver stage nuclear protein (LINUP) (Goswami et al, 2022a). Gene deletion of *LINUP* in both Py and Pf resulted in severe defects in late LS differentiation (Goswami et al, 2022a). Thus, to study the potential for further optimization of attenuation via dual gene deletion, we generated a Py *mei2⁻/linup⁻* double knockout parasite using CRISPR/Cas9-mediated gene editing. Blood stage schizonts of selectable marker-free Py *mei2⁻* (Vaughan et al, 2018) were transfected with a pYC-LINUP plasmid, followed by positive selection with pyrimethamine as previously reported (Vaughan et al, 2018). Recombinant parasites were cloned by limiting dilution and clones were screened by genotyping to confirm deletion of *LINUP* (Appendix Fig. S1A). Two clones, each from independent transfections, were used for further phenotypic analysis. These Py *mei2⁻/linup⁻* double knock-out parasites are referred to herein as PyLARC2. All oligonucleotides used in the study are mentioned in Appendix Table S1.

Mosquito stage infections of PyLARC2 were compared with Py XNL wildtype (PyWT) in *Anopheles stephensi* mosquitoes using three assessments; parasite oocyst counts per infected mosquito midgut (Appendix Fig. S1B), oocyst prevalence in infected blood-fed mosquito populations (Appendix Fig. S1C) and salivary gland sporozoite counts per infected mosquito (Appendix Fig. S1D). This showed that mosquito infections and sporozoite production were comparable between PyLARC2 and PyWT. To determine whether PyLARC2 undergoes complete late LS arrest, PyLARC2 and PyWT salivary gland sporozoites were isolated from mosquitoes and a cohort of 30 highly susceptible BALB/cByJ mice were intravenously (i.v.) challenged with a high dose of 250,000 PyLARC2 sporozoites, while BALB/cByJ challenged with 250,000 PyWT sporozoites served as controls (Table 1). The time to blood stage patency in days was then determined from Giemsa-stained thin blood smears. None of the 30 mice challenged with PyLARC2 sporozoites developed blood stage infections. In contrast, parasites were detected in all three mice challenged with PyWT sporozoites 3 days after challenge. These results indicate that PyLARC2 parasites suffer a complete defect during pre-erythrocytic infection and cannot initiate blood stage infection.

To further investigate PyLARC2 liver infection with focus on late LS development at the cellular level, BALB/cByJ mice were intravenously infected with 250,000 sporozoites of either PyLARC2 or PyWT and mice were euthanized at 36 and 48 h post sporozoite infection (hpi). Infected livers were harvested, fixed and tissue sections were used for immunofluorescence assays (IFA) using parasite-specific antibodies to assess LS development qualitatively and quantitatively. Measurement of LS size revealed no difference in size between PyWT and PyLARC2 LS schizonts at 36 hpi (Fig. 1A). However, at 48 hpi, PyLARC2 late LS schizonts were significantly smaller compared to PyWT (Fig. 1A,B).

A critical event in late LS schizogony is the formation of cytomeres, which are complex invaginations of the parasite plasma membrane (PPM) that significantly increase LS plasma membrane area. These cytomeres are central in the partitioning of the parasite cytoplasm, chromosomes, and organelles and the formation of nascent LS merozoites. To visualize organelle segregation, PyWT and PyLARC2 LS at 48 hpi were labeled with anti-ACP antibodies, which marks the parasite apicoplast. DNA segregation was visualized by DAPI staining. The late LS plasma membrane and LS merozoite plasma membrane was delineated by staining for Py merozoite surface protein 1 (MSP1). PyWT late LS showed MSP1 localization to the cytomeres, which had partitioned the LS cytoplasm (Fig. 1C). The LS apicoplast network displayed well differentiated branches that were in the final stages of segregation (Fig. 1C,D). In addition, multiple punctate DNA centers, indicative of complete DNA segregation, were partitioned along defined cytomere boundaries. In contrast, PyLARC2 late LS schizonts displayed severe abnormalities in differentiation (Fig. 1C,D). They lacked PyMSP-1 expression, indicating a block of LS maturation and LS merozoite formation. The apicoplast remained branched, tubular, and lacked organized distribution (Fig. 1D). Furthermore, PyLARC2 LS showed fewer DNA centers that appeared aggregated, indicating that both DNA replication and segregation were affected. The inner membrane complex protein myosin A tail domain-interacting protein (mTIP) is an additional differentiation marker for LS merozoites. MTIP localized to the inner membrane complex of nascent LS merozoites of PyWT LS at 48 hpi (Fig. 1E). In contrast, no mTIP expression was observed in PyLARC2 LS at 48 hpi, further substantiating that PyLARC2 does not form LS merozoites.

## Immunizations with PyLARC2 GAP protect against pre-erythrocytic infection

To determine the protective efficacy of PyLARC2 immunizations against an infectious Py sporozoite challenge, groups of BALB/cJ mice were intravenously (i.v.) immunized twice with 10,000 PyLARC2 sporozoites 1 month apart. Mice that were mock-immunized with salivary gland extract from uninfected mosquitoes

**Table 1. PyLARC2 exhibits complete liver stage arrest.**

| Parasite genotype | Mouse strain | Number of sporozoites[a] | Number of mice patent/number of mice infected[b] | Days to patency (number of mice)[c] |
|---|---|---|---|---|
| PyWT | BALB/cByJ | 250,000 | 3/3 | 3 (3) |
| PyLARC2 | BALB/cByJ | 250,000 | 0/30 | - |

[a]The number of salivary gland sporozoites intravenously injected per mouse.
[b]The number of patent mice per total mice infected is indicated. Detection of blood stage parasitemia was carried out in Giemsa-stained thin blood smears. Attenuation was considered complete if mice remained blood stage negative for 21 days.
[c]If mice became blood stage patent, the day to patency is listed, with the number of mice that became patent in parentheses.

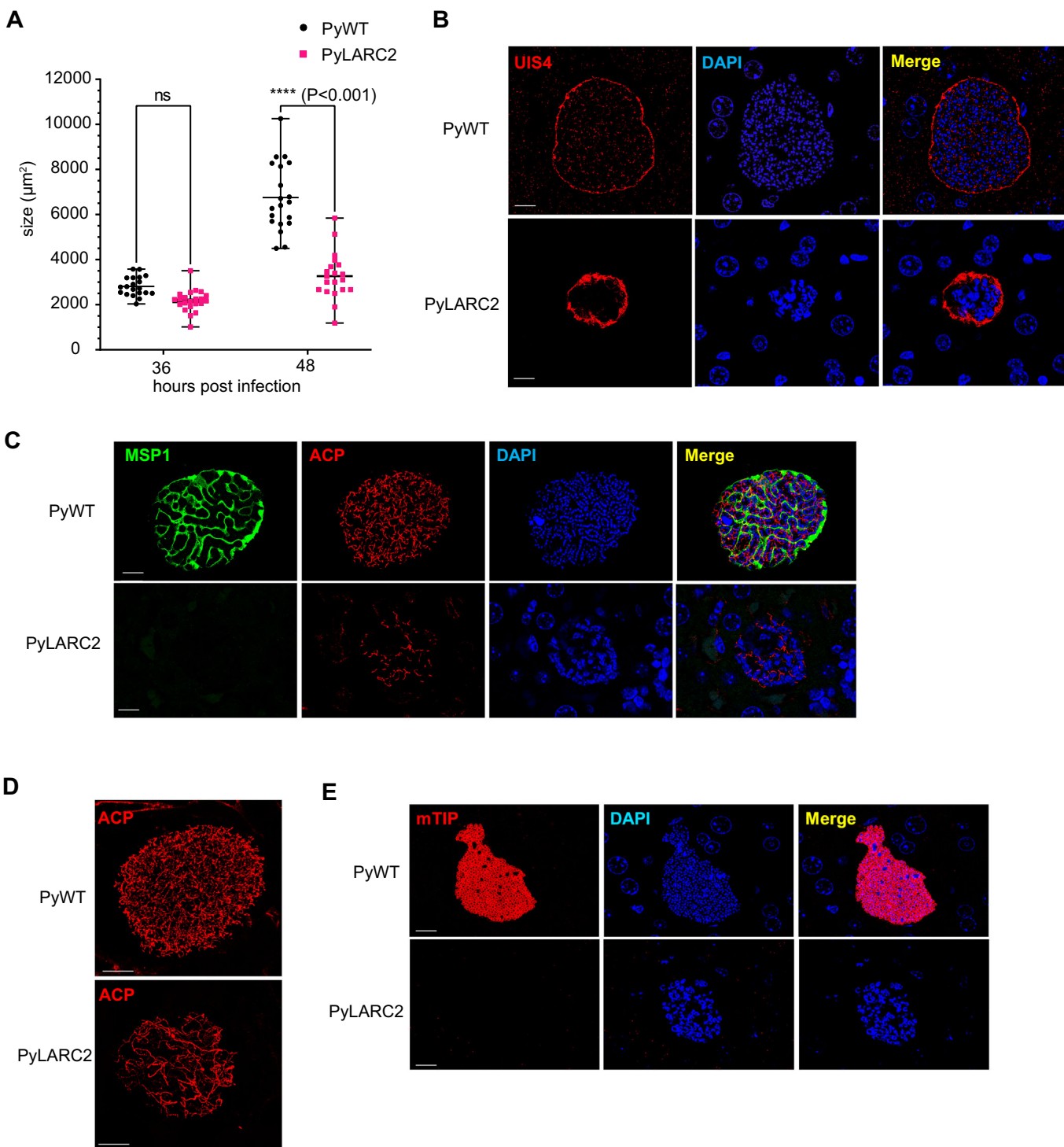

served as a control. One month after the last immunization, mice received an i.v. challenge with 10,000 PyWT sporozoites (Fig. 2A, Table 2). The age matched naive mice that did not receive any immunogen were also included as a control. Mock immunized mice and naive mice developed blood stage infection 3 days after sporozoite challenge (Fig. 2A, Table 2). In contrast, all ten PyLARC2-immunized mice remained blood stage-negative and were thus protected against infectious sporozoite challenge. We

next evaluated PyLARC2 protective efficacy using an intramuscular (i.m.) route of immunization. Groups of BALB/cJ mice were immunized twice with 20,000 PyLARC2 sporozoites a month apart. Mice that were mock-immunized with salivary gland extract from uninfected mosquitoes and naive mice served as a control. All mice then received an i.v. challenge with 10,000 PyWT sporozoites 1 month after the last immunization (Fig. 2B, Table 2). Mock-immunized mice and naive mice developed blood stage infection

◄  **Figure 1.  PyLARC2 shows a complete late liver stage developmental arrest.**

Liver tissue sections were prepared from BALB/cByJ mice infected with 250,000 (250 K) sporozoites of either Py XNL (PyWT) or PyLARC2, 36 and 48 h post infection (hpi) and analyzed by IFA. **(A)** Comparison of LS parasite size (based on area at the LS largest circumference) between PyWT and PyLARC2 at 36 and 48 hpi. This analysis indicates that sizes of replicating LS schizonts at 36 hpi are comparable between PyWT and PyLARC2. However, PyLARC2 late LS schizonts at 48 hpi are significantly smaller than PyWT. Data is represented as mean ± SD. Each datapoint refers to the mean size of at least 20 parasites for each timepoint. Statistical analysis was carried out using two-way ANOVA using Tukey's multiple comparison test. $P$ value is mentioned in the figure legend. $P > 0.05$ is taken as ns. LS development was compared between PyWT and PyLARC2 at 48 hpi using antibodies against; **(B)** the PVM marker UIS4 (red); **(C)** the PPM and mature LS merozoite marker, MSP1 (green), and apicoplast marker, ACP (red); **(D)** the apicoplast marker, ACP (red) with a 3D projection; **(E)** the inner membrane complex and mature LS merozoite marker, mTIP (red). DNA was stained with DAPI (blue). Scale bar is 10 μm. PyLARC2 LS parasites are smaller compared to PyWT and do not express mature merozoite markers MSP1 and mTIP, indicating a lack of LS merozoite formation. Furthermore, organellar and DNA segregation is incomplete in PyLARC2 LS. The apicoplast appears tubular and disorganized and DAPI staining displays large and aggregated DNA centers. In contrast, mature PyWT LS have defined cytomeres visualized with MSP1. The apicoplast and DNA are segregated along cytomere boundaries and in mature LS display complete segregation into LS merozoites. Source data are available online for this figure.

3 days after sporozoite challenge (Fig. 2B, Table 2). Seven of ten mice immunized with PyLARC2 via i.m. route were completely protected against challenge, while the three mice that were blood stage positive displayed significant delays in the onset of blood stage patency, indicating partial protection in these mice.

We next investigated durability of PyLARC2-engendered protection. Groups of BALB/cJ mice were i.v.-immunized twice with 50,000 PyLARC2 sporozoites 1 month apart, or mock immunized and subsequently all mice were i.v.-challenged with 10,000 PyWT sporozoites 6 months (Fig. 2C) or 12 months after the last immunization (Table 2). The age matched naive mice were also included as a control. Mock immunized and naive mice developed blood stage infection 3 days after sporozoite challenge (Fig. 2C, Table 2). Nine of ten mice, and six of ten mice immunized with PyLARC2 were completely protected against challenge at 6 months and 12 months, respectively. We observed a 4- to 7-day delay in time to onset of blood stage patency for the mice that were blood stage positive, indicating partial protection in these mice.

To investigate protection in outbred mice we used Swiss Webster (SW) mice since they were shown to be more difficult to protect by whole sporozoite vaccination compared to inbred BALB/cJ mice (Butler et al, 2011). A cohort of ten mice received three i.v. immunizations of 50,000 PyLARC2 sporozoites at 2-month intervals, followed by an i.v.-challenge with 1000 PyWT sporozoites 2 months after the last boost (Fig. 3A, Table 3). Controls for the experiment included mice that received mock immunizations and the age matched naive mice. All naive and mock-immunized mice developed blood stage infection 4 days after sporozoite infection (Fig. 3A, Table 3). Seven of ten PyLARC2–immunized mice exhibited complete protection against sporozoite challenge, while the three mice that were blood stage positive showed a 1-day delay in the pre-patent period compared to naive and mock infected mice, indicating partial protection in these mice.

We next tested the efficacy of cryopreserved PyLARC2, since all SPZ vaccine products are ultimately used as a cryopreserved vaccine formulations (Hoffman et al, 2010). BALB/cJ mice were immunized twice with 50,000 cryopreserved PyLARC2 sporozoites, 2 weeks apart, followed by i.v.-challenge with 10,000 cryopreserved PyWT sporozoites. All PyLARC2-immunized mice showed complete sterile protection against Py sporozoite challenge, while naive mice that received no immunogen developed blood stage infection within 5 days after sporozoite challenge (Fig. 3B, Table 4).

Together, these results demonstrate that immunizations with PyLARC2 in both inbred and outbred strains of mice engenders sterile protection against pre-erythrocytic stage infection and that

different routes of immunizations can be used for protective vaccination. Furthermore, cryopreserved PyLARC2 sporozoites retained their vaccine potency.

Studies in rodents and in non-human primates have demonstrated that CD8⁺ T cells (Weiss and Jiang, 2012; Schofield et al, 1987; Cockburn et al, 2013; Weiss et al, 1988; Cockburn et al, 2014; Van Braeckel-Budimir and Harty, 2014; Keitany et al, 2014), specifically liver-resident memory CD8⁺ T cells (T$_{RM}$) (Fernandez-Ruiz et al, 2016; McNamara et al, 2017; Olsen et al, 2018), are critical for whole sporozoite vaccine-mediated sterilizing immunity. We therefore characterized CD8⁺ T cells from the liver and spleen of C57BL/6 mice that were immunized with either three doses of 50,000 PyLARC2 sporozoites or were mock-immunized with uninfected mosquito salivary gland extract (Fig. EV1). The mice immunized with PyLARC2 harbored significantly higher numbers of antigen experienced CD44$^{hi}$ CD8⁺ memory T cells in the liver compared to mock control at 2 months after the last immunization (Fig. EV1B i–ii). Importantly, PyLARC2-immunized mice showed two-fold higher CD69$^{hi}$CXCR6$^{hi}$ CD8⁺ T$_{RM}$ cell populations when compared to the mock controls (Fig. EV1C i–iii), indicating that PyLARC2 immunization elicits robust numbers of antigen experienced, liver-resident CD8⁺ T cells. Next we analyzed whether the CD8⁺ memory T cells in the livers of PyLARC2 immunized mice express CD11a, a part of the LFA-1 integrin complex, which is required for the retention of T$_{RM}$ cells in the liver (McNamara et al, 2017). Approximately 80% of the CD44$^{hi}$ CD8⁺ memory T cells in the liver showed high levels of CD11a expression (Fig. EV1D). Indeed, the CD11a$^{hi}$ cells were mostly CD69$^{hi}$CXCR6$^{hi}$ CD8⁺ T$_{RM}$ cells, indicating their potential to be retained in the liver. Moreover, PyLARC2 immunized mice also harbored significantly higher numbers of circulating CD44$^{hi}$ CD62L$^{lo}$CD8⁺ effector memory T cells (T$_{EM}$s) in the spleen (Fig. EV1E i–iii).

LARC2 parasites arrest late in LS development and might express antigens that are shared with blood stages. To investigate whether PyLARC2 pre-erythrocytic stage immunizations elicit blood stage immunity, groups of BALB/cJ mice were i.v.-immunized three times at monthly intervals with 10,000 PyLARC2 sporozoites, while mock-immunized mice received salivary gland extract from uninfected mosquitoes and the age matched naive mice received no immunizations (Fig. 3C). All mice were subsequently challenged i.v. with 10,000 lethal Py XL blood stage parasites, 1 month after the last immunization. Naive and mock-immunized mice were unable to control blood stage parasitemia and had to be euthanized per protocol 7 days after challenge due to high parasite burden. Intriguingly, all mice

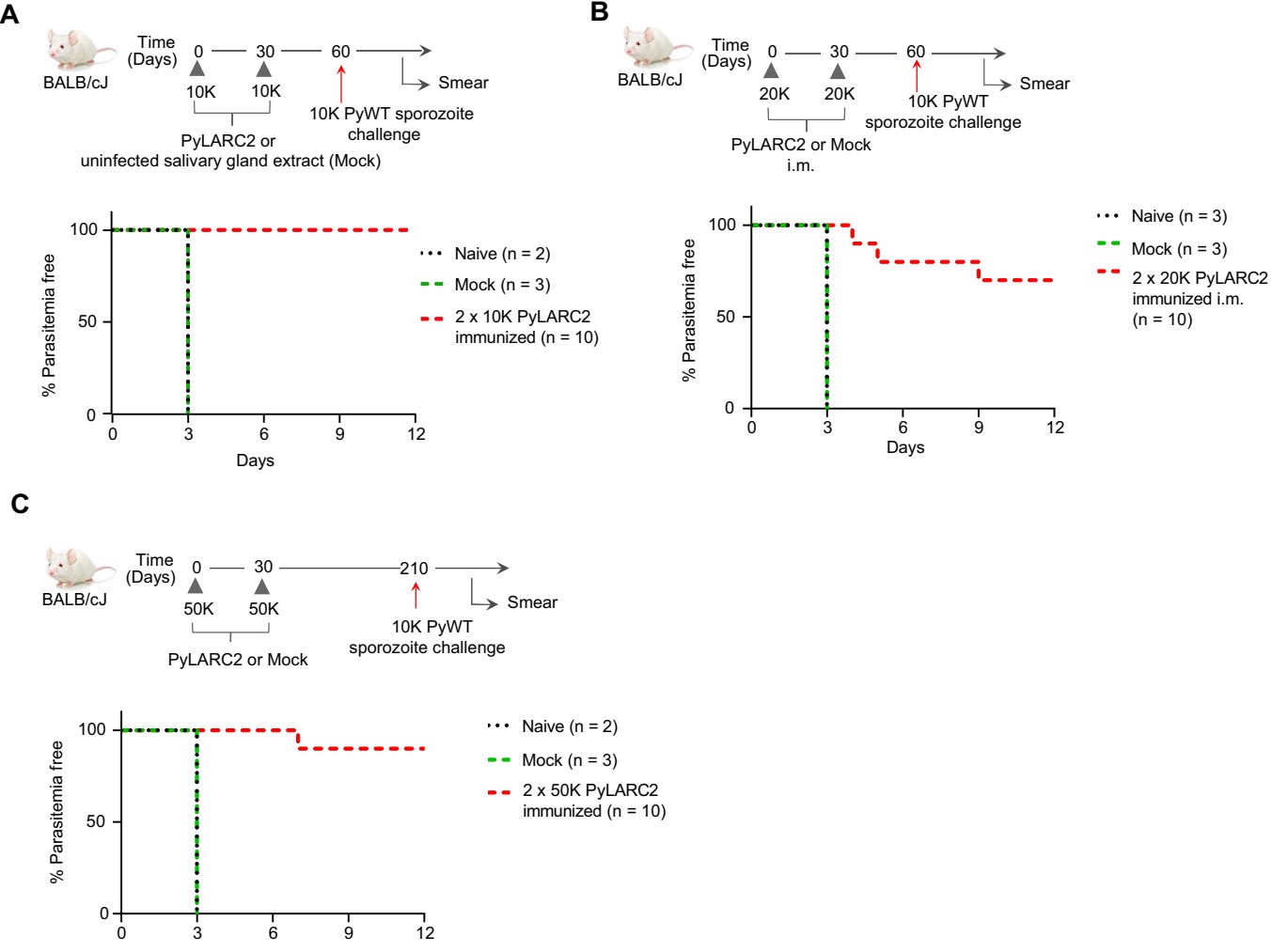

**Figure 2. PyLARC2 immunizations confer durable protection against sporozoite challenge.**

(A) Groups of BALB/cJ mice were immunized and challenged as shown in the schematic. The survival curve shows the percentage of mice that did not show blood stage parasites by thin blood smear, up to 21 days after sporozoite challenge. All ten mice immunized with PyLARC2 were protected after challenge, while both the age matched naive and mock-immunized mice developed blood stage parasitemia 3 days after challenge. (B) Groups of BALB/cJ mice were immunized via the intra-muscular route and challenged as shown in the schematic. The survival curve shows the percentage of mice that did not show blood stage parasites by thin smear up to 21 days after sporozoite challenge. Seven of ten mice immunized with PyLARC2 were protected after challenge, while the three mice that were blood stage positive displayed a 2 to 6 days delay in time to onset of blood stage infection. Both the age matched naive mice and mock-immunized mice developed blood stage parasitemia 3 days after challenge. (C) To determine the durability of PyLARC2-engendered protection, groups of BALB/cJ mice were immunized as shown in the schematic and challenged 6 months after the last boost. The survival curve shows the percentage of mice that did not show blood stage parasites by thin smear, up to 21 days after sporozoite challenge. Nine of ten mice immunized with PyLARC2 were protected after challenge, while the one mouse that developed blood stage parasitemia displayed a four-day delay in time to onset of patency. Both the age matched naive and mock-immunized mice developed blood stage parasitemia three days after challenge. Source data are available online for this figure.

immunized with PyLARC2 had reduced peak parasitemia compared to the control mice and were able to completely clear blood stage parasites 12 days after challenge (Fig. 3C). Plasma was harvested from PyLARC2-immunized BALB/c and SW mice described earlier, 14 days after the second immunization and reactivity of immune sera to PyWT LS and blood stages was evaluated by IFA (Fig. EV2). PyLARC2 immune sera from SW mice recognized PyWT LS at 28, 36, and 48 hpi (Fig. EV2A). At 28 hpi, reactivity against antigens localized to the LS parasitophorous vacuole membrane (PVM) or parasite plasma membrane (PPM) was observed, while for more mature LS, immune sera

recognized both apparent membrane and cytoplasmic antigens. Immune sera from PyLARC2 immunized BALB/c and SW mice also reacted with apparent cytoplasmic antigens of PyWT blood stage parasites (Fig. EV2B). Plasma from mock-immunized mice did not show any reactivity against PyWT LS or blood stage antigens, indicating that the observed reactivity was specific to LARC2-immunized mice (Fig. EV2A,B). These results show that PyLARC2 pre-erythrocytic stage immunizations can protect against a lethal blood stage infection and this might be in part attributable to antibodies elicited against antigens shared between the LS and blood stages.

**Table 2. PyLARC2 immunization protects against pre-erythrocytic stage challenge.**

| Mouse strain | Parasite genotype | Route of immunization | Number of sporozoites used for immunization/challenge | | | Number of mice protected (days to patency)[c] |
|---|---|---|---|---|---|---|
| | | | Prime[a] | Boost (days after prime)[a] | Challenge (days after boost)[b] | |
| BALB/cJ | NA | i.v | – | – | 10,000 (30) | 0/2 (3) |
| | NA | i.v | – | – | 10,000 (180) | 0/2 (3) |
| | NA | i.v | – | – | 10,000 (360) | 0/3 (3) |
| | NA | i.m | – | – | 10,000 (30) | 0/3 (3) |
| | NA | i.v | Salivary gland extract | Salivary gland extract (30) | 10,000 (30) | 0/3 (3) |
| | NA | i.v | Salivary gland extract | Salivary gland extract (30) | 10,000 (180) | 0/3 (3) |
| | NA | i.v | Salivary gland extract | Salivary gland extract (30) | 10,000 (360) | 0/3 (3) |
| | NA | i.m | Salivary gland extract | Salivary gland extract (30) | 10,000 (30) | 0/3 (3) |
| | PyLARC2 | i.v | 10,000 | 10,000 (30) | 10,000 (30) | 10/10 |
| | PyLARC2 | i.v | 50,000 | 50,000 (30) | 10,000 (180) | 9/10 (7) |
| | PyLARC2 | i.v | 50,000 | 50,000 (30) | 10,000 (360) | 6/10 (4, 7) |
| | PyLARC2 | i.m | 20,000 | 20,000 (30) | 10,000 (30) | 7/10 (4, 5, 9) |

[a]PyLARC2 salivary gland sporozoites were isolated from infected *Anopheles stephensi* mosquitoes, and mice were immunized either intravenously (i.v.) or intramuscularly (i.m.) with the listed number of sporozoites. The day intervals after priming are indicated in parentheses.
[b]Mice were i.v. challenged with wild-type salivary gland sporozoites. The day intervals after the boost, when the challenge took place, are indicated in parentheses.
[c]The number of protected mice per total mice challenged. The days to patency are indicated in parentheses. Protection was considered complete if mice remained blood stage negative in Giemsa-stained thin blood smears for 21 days after challenge.

We next conducted experiments to investigate whether PyLARC2 can engender species-transcending pre-erythrocytic immunity. Groups of BALB/cJ mice were i.v.-immunized at monthly intervals with 50,000 PyLARC2 sporozoites, while mock-immunized mice received salivary gland extract from uninfected mosquitoes and the age matched naive mice received no immunogen (Fig. 3D, Table 5). All mice were subsequently i.v.-challenged with 10,000 *P. berghei* ANKA sporozoites, a lethal rodent malaria parasite species that is genetically distinct from Py. All mock-immunized and naive mice developed blood stage infection within 4 to 5 days after sporozoite challenge. All PyLARC2-immunized mice showed complete sterile protection against *P. berghei* sporozoite challenge (Fig. 3B, Table 5). Plasma was harvested from PyLARC2-immunized BALB/c and SW mice 14 days after the second immunization as described above and reactivity of immune sera to *P. berghei* LS and blood stages was evaluated by IFA (Fig. EV3). PyLARC2 immune sera from SW mice recognized both membrane and cytoplasmic antigens of mature *P. berghei* LS (Fig. EV3A), which was similar to what was observed for Py LS (Fig. EV2A). Furthermore, immune sera from PyLARC2-immunized BALB/c and SW mice also reacted with apparent cytoplasmic antigens of *P. berghei* blood stage parasites (Fig. EV3B). Plasma from mock-immunized mice did not show any reactivity against *P. berghei* LS or blood stage antigens, indicating that the observed reactivity was specific to LARC2-immunized mice (Fig. EV3A,B). These results show that PyLARC2-mediated pre-erythrocytic stage immunity can elicit species transcending protection against a *P. berghei* sporozoite challenge and this might be in part attributable to antibodies elicited against antigens shared between the *P. berghei* and Py LS and blood stages.

## Generation of *P. falciparum* LARC2 suitable for human vaccination

Based on the encouraging findings with PyLARC2, we next generated the dual gene deletion human malaria parasite, Pf *mei2⁻/linup⁻* (PfLARC2). We have previously reported the generation of marker-free Pf *mei2⁻* parasites using the pFC-PlasMei2-yFCU plasmid (Fig. 4A) (Goswami et al, 2020). Pf *mei2⁻* clone F3 was transfected with the pYC-LINUP-yFCU plasmid carrying a positive/negative selection cassette, which used a CRISPR/Cas9 expression system to delete the Pf *LINUP* gene (Fig. 4A). Transfected parasites underwent positive selection with the drug WR2210 to select for parasites that had undergone double cross-over recombination. This was followed by two rounds of negative selection with the drug 5-fluorocytosine to eliminate recombinant parasites that still retained copies of the plasmid, as previously described (Goswami et al, 2020). Parasites were screened for the correct recombination pattern indicating *LINUP* deletion using PCR genotyping and recombinant parasite populations were used for cloning by limiting dilution. PfLARC2 clones F7, C12, and B7 were confirmed to carry deletions of *PlasMei2* and *LINUP* genes by PCR genotyping (Fig. 4B–D) and whole genome sequencing (Appendix Fig. S2). In addition, PfLARC2 clones F7, C12, and B7 did not carry any extraneous DNA or any additional deletions and differed from PfNF54-WT only in the deletion of the *PlasMei2* (chromosome 6) and *LINUP* (chromosome 12) genes. All oligonucleotides used in the study are mentioned in Appendix Table S1.

Pf LARC2 clones F7, C12, and B7 were next evaluated for the highest yield of salivary gland sporozoites (PfSPZ), which is critical

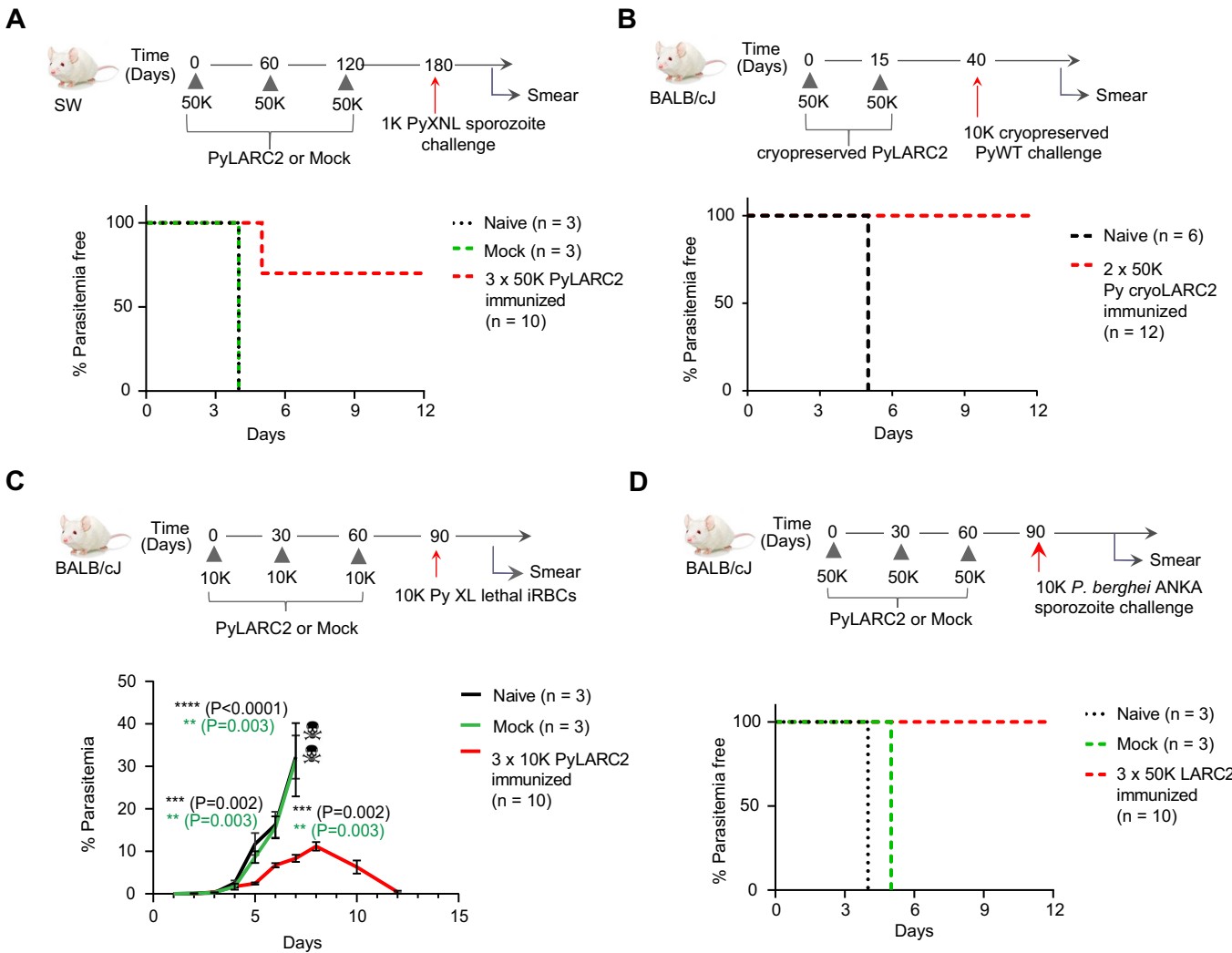

**Figure 3. PyLARC2 immunization confers broad protection against sporozoite challenge and confers protection against blood stage challenge.**

(A) Groups of outbred SW mice were immunized and challenged as shown in the schematic. The survival curve shows the percentage of mice that did not show blood stage parasites by thin blood smear, up to 21 days after sporozoite challenge. Seven of ten mice immunized with PyLARC2 were protected from challenge, while the three mice that developed blood stage parasitemia showed a 1-day delay in time to patency. All the naive and mock-immunized mice developed blood stage parasitemia 4 days after challenge. (B) Groups of BALB/cJ mice were immunized and challenged as shown in the schematic. The survival curve shows the percentage of mice that did not show blood stage parasites by thin blood smear, up to 21 days after sporozoite challenge. All twelve mice immunized with cryopreserved PyLARC2 were protected from challenge. All six naive mice developed blood stage parasitemia 5 days after challenge. (C) Stage transcending protection of PyLARC2 pre-erythrocytic immunization was evaluated as shown in the schematic by immunizing groups of BALB/c mice, followed by an intravenous challenge with 10,000 lethal Py XL infected RBCs (iRBC). All the naive and mock-immunized mice developed parasitemias exceeding 30% after 7 days of iRBC inoculation and had to be euthanized. All ten PyLARC2-immunized mice developed transient blood stage parasitemias up to 10% but cleared blood stage infection 12 days after challenge. Data is shown as mean ± SEM. Statistical analysis was carried out using two-way ANOVA using Tukey's multiple comparison test. Comparison between mock and PyLARC2 immunized mice are shown in green stars and between naive and PyLARC2 immunized mice are shown in black stars. P values are indicated in the figure. P > 0.05 is taken as ns. (D) Protective efficacy of PyLARC2 immunization against a different *Plasmodium* species (*P. berghei*) was evaluated by immunizing groups of BALB/cJ mice as shown in the schematic. The survival curve shows the percentage of mice that did not show blood stage parasites by thin smear, up to 21 days after sporozoite challenge. All ten mice immunized with PyLARC2 were protected after challenge. All the naive and mock-immunized mice developed blood stage parasitemia 4 to 5 days after challenge. Source data are available online for this figure.

for vaccine production. Gametocyte cultures were established using standard protocols and compared to the PfNF54-WT parental wildtype strain. The quality of mature gametocyte cultures was assessed by enumeration of male gametocyte exflagellation for the PfLARC2 clones and PfNF54-WT. This showed similar exflagellation levels between PfNF54-WT and PfLARC2 clones F7 and B7, while clone C12 showed significantly lower exflagellation (Fig. 5A).

Mature gametocytes were then fed to mosquitoes using standard membrane feeding, and mosquito infectivity was analyzed by counting the numbers of oocysts in the midgut (Fig. 5B), oocyst prevalence assessment (Fig. 5C), and enumeration of salivary gland PfSPZ per mosquito (Fig. 5D). The average oocyst prevalence was 81% for PfNF54-WT, and 72%, 52%, and 86% for PfLARC2 clones F7, C12, and B7, respectively (Fig. 5C). PfLARC2 clones F7 and B7

**Table 3. PyLARC2 immunization protects against pre-erythrocytic stage challenge in outbred mice.**

| Mouse strain | Parasite genotype | Route of immunization | Number of sporozoites used for immunization/challenge | | | Number of mice protected (days to patency)[c] |
| | | | Prime[a] | Boost (days after prime)[a] | Challenge (days after last boost)[b] | |
|---|---|---|---|---|---|---|
| SW | NA | i.v. | – | – | 1000 (60) | 0/3 (4) |
| | NA | i.v. | Salivary gland extract | Salivary gland extract (60, 120) | 1000 (60) | 0/3 (4) |
| | PyLARC2 | i.v. | 50,000 | 50,000 (60, 120) | 1000 (60) | 7/10 (5) |

[a]PyLARC2 salivary gland sporozoites were isolated from infected *Anopheles stephensi* mosquitoes, and mice were i.v. immunized with the listed number of sporozoites. The day intervals after priming are indicated in parentheses.
[b]Mice were i.v. challenged with wildtype salivary gland sporozoites. The days after the last boost, when the challenge took place, are indicated in parentheses.
[c]The number of protected mice per total mice challenged. The days to patency are indicated in parentheses. Protection was considered complete if mice remained blood stage negative in Giemsa-stained thin blood smear for 21 days after challenge.

had similar oocyst infection levels in the mosquito midgut compared to PfNF54-WT but C12 had significantly lower numbers of oocysts (Fig. 5B). Among the PfLARC2 clones, clone F7 produced the highest numbers of salivary gland PfSPZ, with an average of 30,000 PfSPZ per mosquito, comparable to PfNF54-WT, which had an average of 45,000 PfSPZ per mosquito (Fig. 5D). Therefore, PfLARC2 clone F7 was selected for further evaluation.

To test PfSPZ-LARC2 production in the Sanaria sporozoite manufacturing facility, mosquito infectivity of PfLARC2 clone F7 was evaluated under aseptic conditions. Research-grade (non-GMP) PfSPZ-LARC2 were produced from aseptic mosquitoes using aseptic processes. Gametocytogenesis for PfLARC2 clone F7 was induced by limiting fresh erythrocytes, and mature stage V gametocytes were fed to aseptic female *Anopheles stephensi* mosquitoes following sterile procedures by membrane feeding. PfSPZ-LARC2 clone F7 exhibited an average oocyst prevalence of 85%, an average of 57 oocysts/midgut, and generated an average of $1.06 \times 10^5$ PfSPZ/mosquito (Fig. 5E), indicating robust sporozoite production. Mosquitoes were then dissected, and PfSPZ-LARC2 were purified, cryopreserved, and vialed following Sanaria's methods for PfSPZ research products, and stored in liquid nitrogen vapor phase at below −150 °C at a concentration of $9 \times 10^5$ PfSPZ per vial on day 16 post infectious feed (Hoffman et al, 2010).

## PfLARC2 shows late liver stage arrest and complete attenuation in humanized FRG huHep mice

To evaluate LS development of PfLARC2 parasites, three million aseptic, purified, and cryopreserved PfSPZ-LARC2 or wild-type NF54 PfSPZ (PfSPZ-WT) were injected i.v. into two FRG NOD huHep mice per group, respectively (Fig. 6A). Mice were euthanized 6 and 7 days post injection. Livers were harvested, fixed and were subjected to IFA using parasite-specific antibodies to examine LS burden, LS growth and cellular differentiation by microscopy. Liver stage burden was compared by counting the number of LS parasites per cm² area of liver tissue with no significant differences observed between PfPSZ-WT and PfSPZ-LARC2 6 days after sporozoite injection (Fig. 6B). Late LS growth was compared between PfSPZ-WT and PfSPZ-LARC2 by measuring the size of LS parasites on days 6 and 7 of development. This showed that LS size was comparable between PfSPZ-WT and PfSPZ-LARC2 (Fig. 6C).

Next, cellular differentiation was compared between PfSPZ-WT and PfSPZ-LARC2 using parasite-specific antibodies. PfSPZ-WT

LS schizonts displayed invaginating cytomeres, marked by CSP staining, on 6 (Fig. EV4B) and 7 days (Fig. 6D) post infection, which preceded segregation of the LS cytoplasm and culminated in the formation of liver stage merozoites, visualized with MSP1 staining at 7 days of LS development. In contrast, at 6 days of development, PfSPZ-LARC2 LS displayed an atypical CSP distribution pattern that indicated formation of new PPM components, but it lacked organization into defined cytomeres (Fig. EV4B). By 7 days of development, PfSPZ-LARC2 LS had ceased to undergo further differentiation and showed no evidence that mature LS merozoites had formed (Fig. 6D).

To specifically visualize LS merozoites, IFAs were performed on infected liver sections from 7 days of development using antibodies against PfMSP1 and mTIP. PfSPZ-WT LS showed Pf MSP1 (Fig. 6E) and mTIP (Fig. 6F) expression and a localization to numerous nascent LS merozoites. In contrast, PfSPZ-LARC2 LS showed weak and disorganized Pf MSP-1 expression (Fig. 6E) and there was no apparent expression of mTIP (Fig. 6F). This indicated a lack of LS merozoite formation. Completion of LS schizogony also involves segregation of replicated organelles and DNA and their partitioning into LS merozoites. To visualize and compare the segregation of organelles and DNA between PfSPZ-WT and PfSPZ-LARC2, sections from infected livers at 7 days of development were used for IFA with antibodies against mitochondrial HSP70 (mHSP70) and the apicoplast protein ACP. DAPI was used to visualize DNA. PfSPZ-WT LS schizont mitochondria and apicoplasts were visible as numerous small spherical structures that clustered with segregated DNA arranged in circular structures (Fig. EV4C). In contrast, PfSPZ-LARC2 LS schizonts contained mitochondria and apicoplasts that appeared tubular and disorganized. The PfSPZ-LARC2 LS schizont DNA also appeared disorganized and arranged in small clusters, which was strikingly distinct to the organized DNA structures observed in PfSPZ-WT LS schizonts (Fig. EV4C).

Although the comparative analysis of cellular differentiation indicated a complete defect in PfSPZ-LARC2 LS merozoite differentiation, this method is not sufficiently sensitive to ascertain a complete block of LS merozoite formation and their transition to blood stage infection. To capture potential PfSPZ-LARC2 LS-to-blood stage transition, one million PfSPZ-WT and one million PfSPZ-LARC2 sporozoites were injected intravenously into four and six FRG NOD huHep mice, respectively (experimental scheme, Fig. 7A). Mice were injected intravenously with human red blood cells (RBC), 6 and 7 days post infection, to capture viable LS

**Table 4. Immunization with cryopreserved Py LARC2 protects against pre-erythrocytic stage challenge.**

| Mouse strain | Parasite genotype | Route of immunization | Number of sporozoites used for immunization/ challenge | | | Number of mice protected (days to patency)[c] |
| | | | Prime[a] | Boost (days after prime)[a] | Challenge (days after boost)[b] | |
|---|---|---|---|---|---|---|
| BALB/c | NA | NA | – | – | 10,000 (25) | 0/6 (5) |
| | Cryopreserved PySPZ-LARC2 | i.v. | 50,000 | 50,000 (15) | 10,000 (25) | 12/12 |

[a]Mice were i.v. immunized with 50,000 cryopreserved PyLARC2 sporozoites. The day interval between immunizations is indicated in parentheses.
[b]Mice were i.v. challenged with 10,000 cryopreserved wild-type salivary gland sporozoites. The days after the boost, when the challenge took place, are indicated in parentheses.
[c]The number of protected mice per total mice challenged. The days to patency are indicated in parentheses. Protection was considered complete if mice remained blood stage negative in Giemsa-stained thin blood smear for 21 days after challenge.

merozoites capable of RBC infection and the initiation of the blood stage phase (Foquet et al, 2018; Vaughan et al, 2012; Goswami et al, 2020, 2022b). FRG NOD huHep mice were euthanized four hours after the second human RBC injection on day 7 post infection. Blood was collected by cardiac puncture and 50 µl of blood from each mouse was used for qRT-PCR analysis to detect parasite 18S RNA (Pf 18S qRT-PCR) (Seilie et al, 2019). The assay can detect as few as one ring stage parasite equivalent in 50 µl of blood sample or 20 parasite equivalents/ml, assuming a conversion factor of 7400 copies of 18S rRNA per ring-stage parasite. The remaining blood was washed and transferred to in vitro parasite blood culture. Fresh media was replaced daily, and cultures were analyzed every 2 to 3 days for the presence of parasites by Giemsa-stained blood culture smears for up to 6 weeks. After 7 days on in vitro culture (14 days after PfSPZ infection), 50 µl blood was also analyzed by qPCR for the presence of parasites. Livers of infected FRG NOD huHep mice were also harvested 7 days post infection for RNA extraction to analyze LS burden by Pf 18S qRT-PCR. On day 7, blood of PfSPZ-WT-infected FRG NOD huHep mice showed parasite densities between $10^3$–$10^6$ parasite equivalents/ml by Pf 18S qRT-PCR. Parasite densities increased to ~$10^8$–$10^9$ parasite equivalents/ml after 7 days of in vitro culture (14 days after PfSPZ-WT sporozoite infection (Fig. 7B, Table 6; Appendix Table S2), indicating robust blood stage replication and increases in parasite biomass. In addition, blood stage parasites were detected by Giemsa-stained smears in all blood cultures from the four PfSPZ-WT-infected FRG NOD huHep mice within 1 to 3 days of transition from LS to blood stage (Table 6). In PfSPZ-LARC2-infected FRG NOD huHep mice, 0–$2.6 \times 10^3$ parasite equivalents/ml were detected by Pf 18S qRT-PCR in the blood (Appendix Table S2) on day 7. Strikingly, however, after 7 days in culture there was a drastic decline in Pf 18S rRNA signal to 0–58 parasite equivalents/ml. This demonstrated that no viable RBC-infectious LS merozoites were released from PfSPZ-LARC2 LS schizonts (Fig. 7B, Table 6; Appendix Table S2). Furthermore, no PfSPZ-LARC2 blood stage parasites were detected in any blood culture from all six infected FRG NOD huHep mice for the entire 6-week period of culture, as determined by Giemsa-stained thin culture smears.

LS burden after 7 days of development was comparable between PfSPZ-WT- and PfSPZ-LARC2-infected FRG NOD huHep mice when measured by Pf 18S qRT-PCR (Fig. 7C), indicating that the lack of LS-to-blood stage transition observed for PfSPZ-LARC2 was not due to a lack of liver infection or a defect in LS growth,

which agreed with the microscopic evaluation of LS burden (Fig. 6B,C).

Together, these findings demonstrate that PfSPZ-LARC2 LS can grow, replicate, and reach cell sizes comparable to PfSPZ-WT but in contrast to WT, PfLARC2 LS do not form infectious merozoites, and are incapable of initiating blood stage infection. Thus, PfSPZ-LARC2 is completely attenuated in the final phase of LS development.

## Discussion

Pre-erythrocytic stage Pf malaria vaccines that target the sporozoites and LS-infected hepatocytes can prevent the onset of blood stage infection, abrogate disease transmission, and could contribute to regional Pf malaria elimination and global Pf malaria eradication. Over the last decade, clinical trials with radiation-attenuated PfSPZ (PfSPZ Vaccine), administered by syringe, have shown that this vaccine is safe, well tolerated and protective against homologous and heterologous CHMI in malaria-naive subjects in the US and Europe as well as against natural infection in malaria-exposed adults in Africa (Epstein et al, 2011; Ishizuka et al, 2016; Jongo et al, 2018, 2019, 2021; Oneko et al, 2021; Sissoko et al, 2022, 2017; Epstein et al, 2017; Lyke et al, 2017). However, the PfSPZ in PfSPZ Vaccine suffer liabilities, as they do not replicate due to radiation-induced DNA damage, arrest early in LS development and immunizations require three doses of $9 \times 10^5$ PfSPZ to elicit robust sterilizing immunity (Sissoko et al, 2022). In contrast, fully infectious PfSPZ administered in combination with the blood stage antimalarial drug chloroquine, PfSPZ-CVac (CQ), have shown increased efficacy, durability and breadth of protection at much lower doses. Three doses of $1.1 \times 10^5$ PfSPZ of PfSPZ-CVac (CQ) (49) (Mordmüller et al, 2017; Mwakingwe-Omari et al, 2021) gave similar efficacy as three doses of $9 \times 10^5$ PfSPZ of PfSPZ Vaccine (51). This superior performance has been attributed to the increase in parasite antigen biomass and diversity during the dramatic LS growth and replication phase in hepatocytes (Zanghi et al, 2023). However, due to the safety concerns associated with injecting non-attenuated parasites and having to rely on a co-administered drug for safety, the imperative for PfSPZ-CVac product development has diminished. Thus, LARC GAPs represent the best solution for a safe and potent pre-erythrocytic stage vaccine as they are intrinsically attenuated, unfold their full growth potential during

**Table 5. PyLARC2 immunization protects against *P. berghei* pre-erythrocytic stage challenge.**

| Mouse strain | Parasite genotype | Route of immunization | Number of sporozoites used for immunization/challenge | | | Number of mice protected (days to patency)[c] |
|---|---|---|---|---|---|---|
| | | | Prime[a] | Boost (days after prime)[a] | Challenge (days after last boost)[b] | |
| BALB/cJ | NA | i.v | – | – | 10,000 *P. berghei* (30) | 0/3 (4) |
| | NA | i.v | Salivary gland extract | Salivary gland extract (30, 60) | 10,000 *P. berghei* (30) | 0/3 (5) |
| | PyLARC2 | i.v | 50,000 | 50,000 (30, 60) | 10,000 *P. berghei* (30) | 10/10 |

[a]PyLARC2 salivary gland sporozoites were isolated from infected *Anopheles stephensi* mosquitoes, and mice were i.v. immunized with the listed number of sporozoites. The day intervals after priming are indicated in parentheses.
[b]Mice were challenged intravenously with *P. berghei* ANKA salivary gland sporozoites. The days after the last boost, when the challenge took place, are indicated in parentheses.
[c]The number of protected mice per total mice challenged. The days to patency are indicated in parentheses. Protection was considered complete if mice remained blood stage negative in Giemsa-stained thin blood smear for 21 days after challenge.

hepatocyte infection but do not require chemoprophylaxis to prevent blood stage infection. We report herein, the generation of a fully attenuated PfLARC2 GAP vaccine strain that undergoes almost complete LS development but then suffers late LS developmental arrest and cannot transition to blood stage infection. We further show that the rodent malaria LARC2 GAP confers broad and durable sterile protection.

A key feature of a LARC GAP intended for use in vaccination is that the whole parasite immunogen undergoes near full LS development but suffers complete attenuation before transition to disease-causing blood stage infection. In previous studies with rodent malaria models, single gene deletion of a type II fatty acid biosynthesis enzyme (Py *fabb/f*⁻) resulted in late LS attenuation (Butler et al, 2011). Unfortunately, deletion of the orthologous gene in Pf resulted in a discordant phenotype in that it caused a severe defect in sporogony (Vaughan et al, 2009). Thus we focused our efforts on *PlasMei2* (Dankwa et al, 2016) but previously found that although Py *mei2*⁻ displayed severe attenuation of late LS development (Dankwa et al, 2016), it caused rare blood stage breakthroughs when highly susceptible BALB/cByJ mice were infected with a high dose of Py *mei2*⁻ sporozoites. Deletion of *PlasMei2* in Pf showed a phenotype comparable to Py *mei2*⁻ in that it resulted in severe LS attenuation in the FRG huHep humanized mouse model (Goswami et al, 2020). To mitigate concerns that the Pf *mei2*⁻ GAP is incompletely attenuated, based on instances of rare blood stage breakthroughs observed with Py *mei2*⁻ parasites, we continued our quest to identify novel gene candidates for generating late LS-attenuated parasite strains, with the prospect that combined gene deletions would generate a completely attenuated LARC GAP. A small gene deletion screen recently conducted by us identified a gene called *LINUP*, which is expressed in the later phases of LS development. Deletion of *LINUP* in both Py and Pf resulted in severe but incomplete attenuation of LS development (Goswami et al, 2022a). Therefore, we combined deletions of the *PlasMei2* and *LINUP* genes in both the Py and Pf parasites to generate LARC2 versions for each species. Complete LS attenuation of PyLARC2 was established herein by showing the absence of blood stage breakthroughs in highly susceptible BALB/cByJ mice that were challenged with 250,000 PyLARC2 sporozoites. The same dose of sporozoites had resulted in rare blood stage breakthroughs with single gene deletion Py *mei2*⁻ parasites (Vaughan et al, 2018) and a five-fold lower sporozoite dose had resulted in 45% blood stage breakthroughs with the single gene

deletion Py *linup*⁻ parasites (Goswami et al, 2022a). This indicates that the simultaneous loss of *PlasMei2* and *LINUP* caused a synthetic lethal phenotype, a phenomenon well described in oncology (O'Neil et al, 2017), leading to complete developmental arrest of LARC2 late stage LS parasites. LINUP localizes with LS DNA segregation centers and shows partial co-localization with regions of open chromatin, suggesting a role in regulating gene expression (Goswami et al, 2022a). In contrast, PlasMei2 is a putative RNA binding protein with a cytoplasmic granular localization pattern and might play a post-transcriptional role in regulating LS transcript stability and homeostasis. Both proteins have a similar onset of expression during mid to late LS schizogony. Therefore, it is possible that LINUP and PlasMei2 act in concert to regulate gene expression at the transcriptional and posttranscriptional level, respectively. Notwithstanding, the precise genetic and phenotypic interactions of simultaneous loss of *Mei2* and *LINUP* during LS schizogony remain to be elucidated.

Cellular analysis of LS development showed that both Py and PfLARC2 undergo LS schizogony including significant cell expansion, extensive organelle replication and DNA replication. However, late differentiation events that lead to completion of LS schizogony were severely impaired in LARC2 parasites. We observed several differentiation defects that were conserved in both Py and PfLARC2 LS schizonts at equivalent developmental time points of 48 h pi and at 7 days post infection, respectively. First, segregation of organelles and DNA were severely impaired. Second, LARC2 lacked pronounced cytomeres; extensive LS plasma membrane invaginations that precede formation of LS merozoites. However, there were significant differences in LS cell size expansion between Py and PfLARC2 when compared to respective WT LS. PyLARC2 late LS were significantly smaller than PyWT and displayed growth arrest between 36–48 of LS development. Of note, the LS growth defect of PyLARC2 was phenotypically more similar to that of Py *linup*⁻ (Goswami et al, 2022a) parasites rather than the Py *mei2*⁻ single gene deletion parasites (Dankwa et al, 2016), as the LS size of the latter was comparable to WT. In contrast, the LS size of PfLARC2 and Pf WT were comparable, although we have previously shown that Pf *linup*⁻ late LS schizonts were smaller than WT (Goswami et al, 2022a). It is possible that the loss of both gene products severely dis-regulated the processes that coordinate LS differentiation and cell growth. Unfortunately, precisely delineating the cellular perturbations caused by single versus dual gene deletions is currently very challenging as there is a paucity of

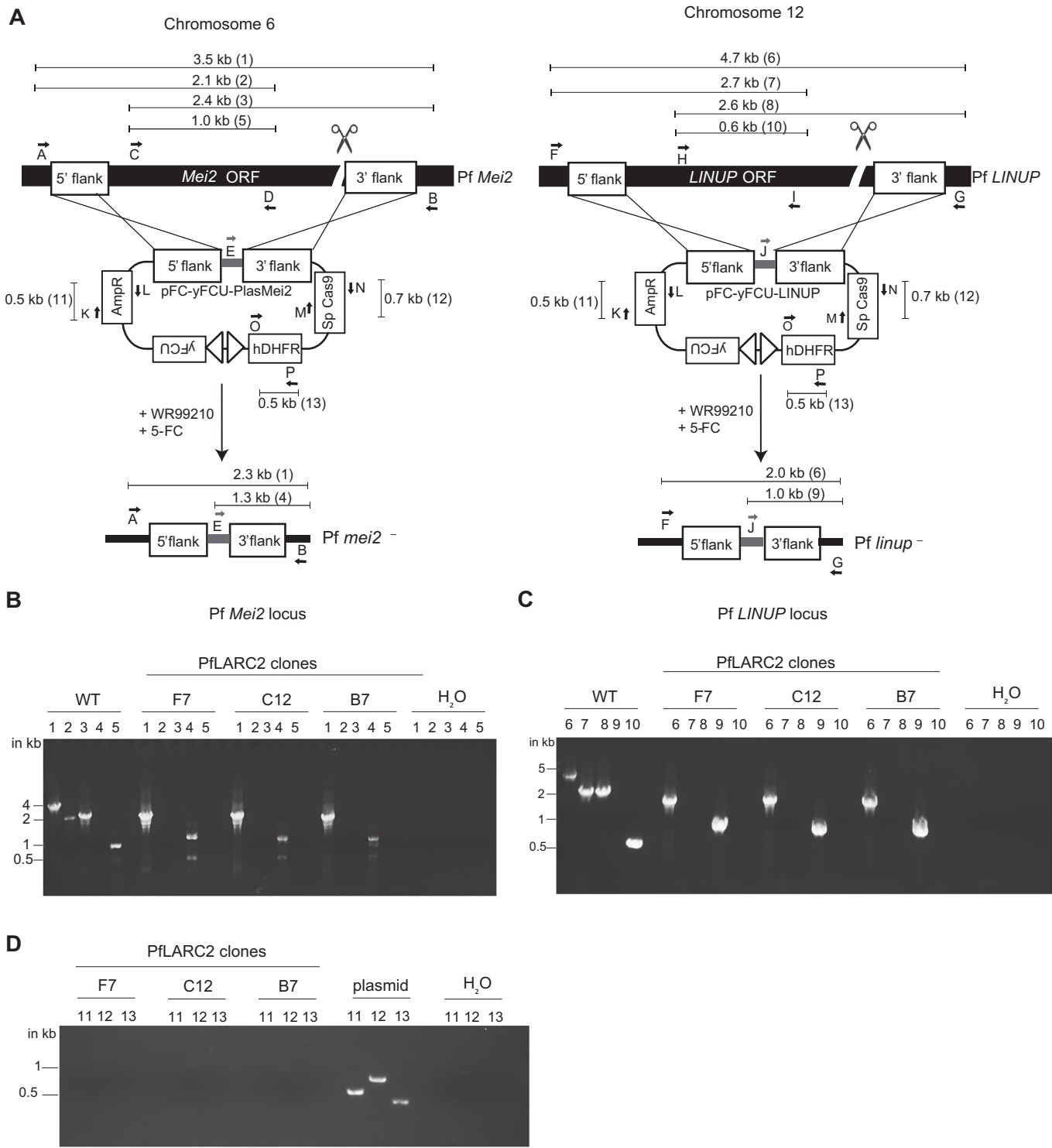

**Figure 4. Generation of *P. falciparum mei2⁻/linup⁻* dual gene deletion strain (PfLARC2).**

(**A**) PfLARC2 was created by first deleting Pf *Mei2* on chromosome 6 and then Pf *LINUP* on chromosome 12 using plasmids pFC-yFCU-Plasmei2 and pFC-yFCU-LINUP, respectively. Sequential positive and negative selection with WR99210 and 5FC resulted in marker-free PfLARC2 free of extraneous DNA. Primers used for confirming parasite genotypes are shown as letters. Expected amplicon sizes for WT and PfLARC2 and the pFC plasmid are indicated in the schematic. Agarose gel electrophoresis to confirm, (**B**) gene deletion of *PlasMei2*, (**C**) gene deletion of *LINUP*, and (**D**) lack of plasmid in three PfLARC2 clones F7, C12, and B7, respectively. Source data are available online for this figure.

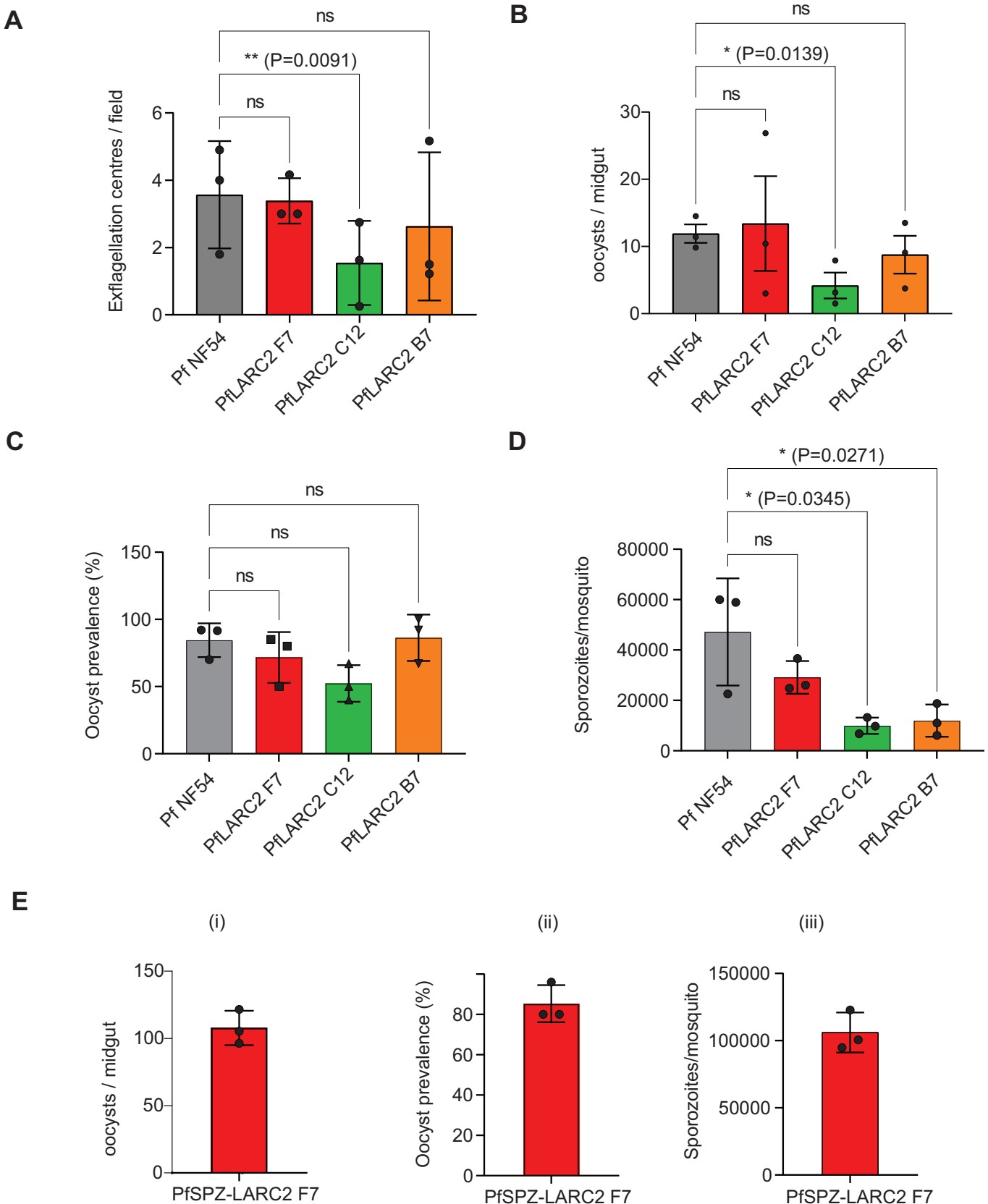

◄

**Figure 5. Analysis of PfLARC2 mosquito stage parasite infection and sporozoite production.**

(A) Exflagellation centers were counted in mature gametocyte cultures of PfLARC2 on day 15 post set up for three PfLARC2 clones F7 (red), C12 (green), B7 (orange), and found to be comparable to PfNF54-WT (gray). These gametocyte cultures were fed to female *Anopheles stephensi* mosquitoes in standard membrane feeding assays and midguts were dissected on day 8 post feed. (B) Oocyst prevalence and, (C) counts for oocyst/mosquito and (D) sporozoites/mosquito were comparable to PfNF54-WT for PfLARC2 for clone F7, therefore clone F7 was used for all further phenotypic evaluation. (E) Research grade (non-GMP) sporozoites were generated from PfLARC2 clone F7 (PfSPZ-LARC2) by feeding mature gametocytes from PfLARC2 clone F7 to aseptic mosquitoes following aseptic procedures. Robust infection of mosquitoes was confirmed by (i) numbers of oocysts/midgut, (ii) oocyst prevalence and (iii) salivary gland sporozoites/mosquito. In (A-D), data is represented as mean $+/-$ SD, $n = 3$ biological replicates. Statistical analysis was carried out using two-way ANOVA using Tukey's multiple comparison test. P values are indicated in the figure. $P > 0.05$ is taken as not significant. In (E), data is represented as mean $+/-$ SD, $n = 3$ biological replicates. For quantification of oocyst/midgut and oocyst prevalence, at least 12 mosquitoes were dissected per strain, and for quantification of sporozoites/mosquito, at least 50 mosquitoes were dissected per strain. Source data are available online for this figure.

knowledge regarding the molecular mechanisms that control LS development. For instance, the molecular networks that coordinate the initiation of cytomere formation once a certain LS cell mass is reached remain completely unknown. Furthermore, it is not understood how DNA and organelle replication is coordinated with the differentiation of merozoites, which is magnitudes greater in LS schizogony when compared to intra-erythrocytic schizogony (Roques et al, 2023). LARC GAPs may serve as useful tools to address some of these fundamental knowledge gaps in the future.

LARC GAP strains are designed with the intent to elicit potent protective immune responses against pre-erythrocytic infection. Due to a lack of small animal models to test the efficacy of Pf GAPs, we assessed the immunogenicity and efficacy of the PyLARC2 GAP in mice. Two i.v. doses of 10,000 PyLARC2 sporozoites conferred complete sterile protection against a high dose i.v. challenge with infectious sporozoites. Importantly, we observed that i.m. immunization with two doses of 20,000 PyLARC2 sporozoites elicited 70% sterilizing protection against high dose i.v. sporozoite challenge, a result that cannot be achieved with similar does of EARD vaccines (Ishizuka et al, 2016; Kramer and Vanderberg, 1975; Douradinha et al, 2007; Patel et al, 2019). It has been shown that i.m. administered sporozoites home to the liver less effectively, reducing the hepatic immunogen load that primes protective CD8$^+$ T cell responses (Ploemen et al, 2013; Silvestre et al, 1970; Patel et al, 2019). PyLARC2 LS replicate and undergo massive cell expansion, which in turn increases antigen biomass within the infected hepatocytes. This might in part explain the good efficacy observed herein with LARC2 i.m. vaccination. The data thus provide substantial evidence to further explore the opportunity for PfSPZ-LARC2 immunizations in humans through the i.m. route. Furthermore, we observed that PyLARC2 immunization induced robust protection in outbred SW mice, which are inherently difficult to protect. We also demonstrate that PyLARC2 confers protection against a heterologous species *P. berghei* sporozoite challenge. Although the antigen-specific targets of LS immunity are largely unknown, these results indicate that the antigens expressed during *Plasmodium* LS development might be conserved across malaria parasite species and this might provide species-transcending protection (Sedegah et al, 2007). Finally, we show that vaccination of mice with cryopreserved PySPZ-LARC2 induces sterilizing immunity against sporozoite challenge. This is important as any whole sporozoite vaccine must currently be cryopreserved to retain efficacy of the live-attenuated sporozoite immunogen.

Whole sporozoite immunization-mediated protection is attributable to both humoral and cellular immune responses. However, studies from rodent as well as non-human primate models have demonstrated that whole sporozoite vaccine-engendered sterilizing immunity is strictly dependent on CD8$^+$ T cells (Bijker et al, 2013; Van Braeckel-Budimir and Harty, 2014; Holz et al, 2016; Weiss and Jiang, 2012; Weiss et al, 1988). In fact, CD8$^+$ T cells alone are capable of conferring sterile protection in the absence of antibody responses (Mueller et al, 2007), indicating their essential role in LS immunity. In addition to protective CD8$^+$ T$_{RM}$ cells that patrol liver sinusoids and reside in the liver, recent work identified CD8$^+$ T$_{EM}$ cells that contribute to protection against LS infection (Fernandez-Ruiz et al, 2016; Vaughan et al, 2018; Lefebvre et al, 2021). We herein observed the presence of a large proportion of CD44$^{hi}$ CD69$^{hi}$ CXCR6$^{hi}$ T$_{RM}$ cells in the liver and CD44$^{hi}$CD62L$^{lo}$ T$_{EM}$ cells in the spleen following PyLARC2 immunization. We further observed that CD8$^+$ T$_{RM}$ cells express high levels of CD11a, which is a part of LFA-1 integrin complex that is required for the retention of CD8$^+$ T$_{RM}$ cells in the liver and for protection against LS infection (McNamara et al, 2017). Interestingly, we observed that a subset of splenic CD8$^+$ T$_{EM}$ cells from PyLARC2 immunized mice also expressed high levels of CD11a, indicating their potential to be retained in the liver following homing.

Our data further show that PyLARC2 GAP confers stage-transcending protection. Mice immunized with PyLARC2 GAP cleared and survived a lethal Py blood stage challenge, indicating that a substantial number of antigens might be shared between PyLARC2 LS and blood stage parasites (Sack et al, 2015). In support of this notion, we show that immune sera from LARC2-immunized mice reacted with late LS and blood stages. However, whether this humoral stage-transcending reactivity contributes to protection against blood stages remains unknown. Interestingly, we have previously observed that Balb/c mice immunized with the *Py fabb/f*$^{-/-}$ LARC GAP loose protection to lethal Py blood stage challenge upon depletion of CD4/CD8$^+$ T cells, but T cells did not substantially contribute to protect C57BL/6 mice, in which protection was largely antibody-dependent (Sack et al, 2015).

The next step in PfSPZ-LARC2 vaccine development is the production of sporozoites suitable for human use. We thus produced aseptic, purified, vialed, cryopreserved PfSPZ-LARC2 using the same methods that have been used to produce Sanaria® PfSPZ Vaccine, PfSPZ Challenge, and PfSPZ-GA1 for clinical trials (Roestenberg et al, 2013; Hoffman et al, 2010; Epstein et al, 2011; Mordmüller et al, 2017; Roestenberg et al, 2020). These PfSPZ-LARC2 vaccine preparations were then used to ascertain complete lack of viable blood stage breakthrough infection using the FRG NOD huHep mouse model reconstituted with human RBCs. This demonstrated that i.v. injection of one million PfSPZ-LARC2 did not lead to viable breakthrough blood stage infection. However, we could detect parasite nucleic acid in the blood of the PfSPZ-

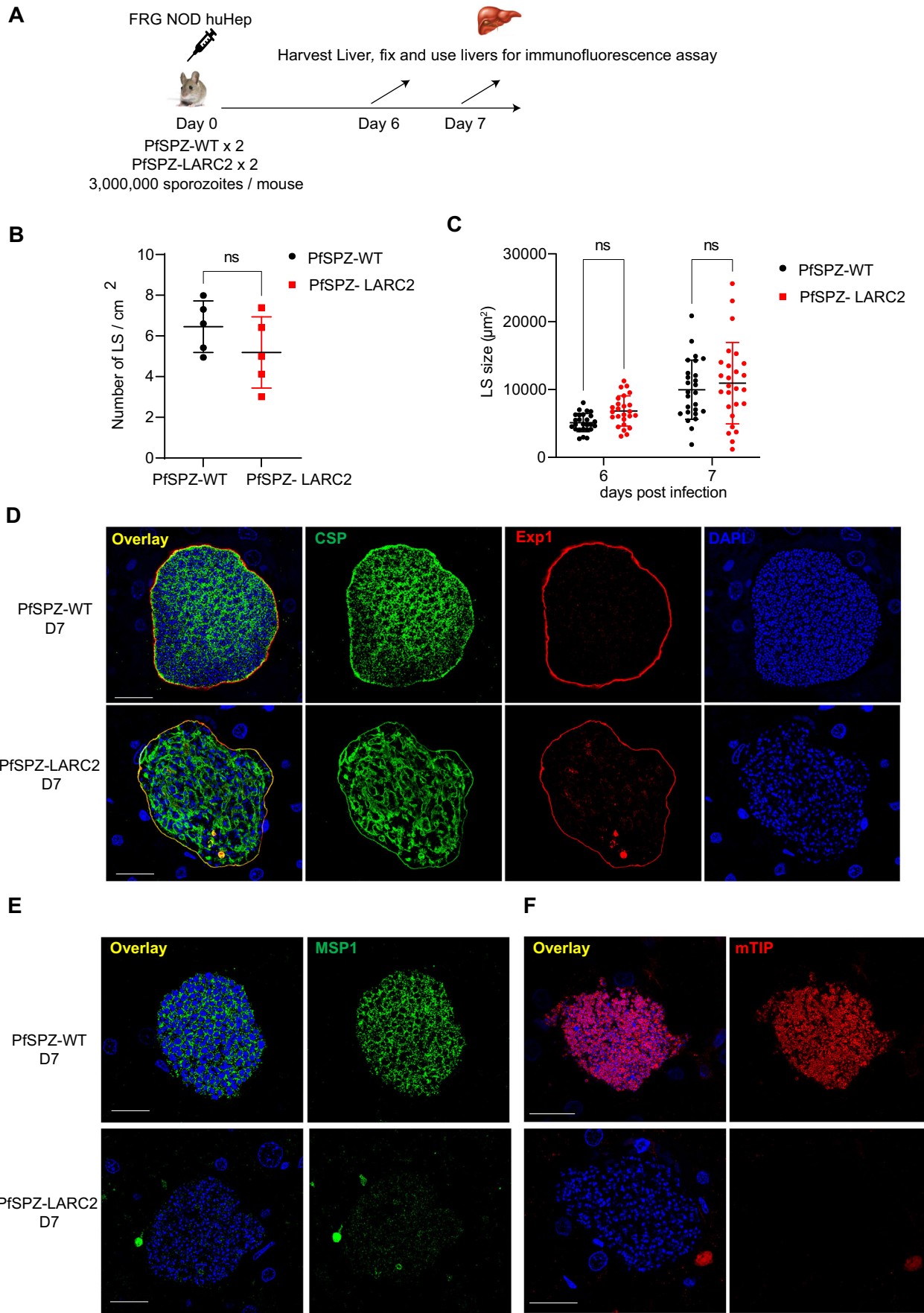

◄

**Figure 6. PfLARC2 parasites display severe defects in late liver stage differentiation.**

(A) The schematic depicts the experimental design. To evaluate LS development of PfLARC2 in FRG NOD huHep mice, three million aseptic, cryopreserved sporozoites of either PfNF54-WT parent strain (PfSPZ-WT) and three million PfSPZ-LARC2 were injected intravenously into two FRG NOD huHep mice per group, respectively. Livers were harvested on days 6 and 7 post infection, fixed and liver tissue sections used for IFA analysis. (B) Comparison of PfSPZ-WT and PfSPZ-LARC2 LS parasite infection density at 6 days post infection. Liver stages were counted from sections of four liver lobes of mice infected with either PfSPZ-WT or PfSPZ-LARC2 within an approximate total area of 500 mm$^2$ of tissue using CSP and Exp1 antibodies as parasite markers. There is no significant difference in LS infection density between PfSPZ-WT and PfSPZ-LARC2. Data is represented as mean ± SD. Each datapoint refers to the number of LS parasites per cm$^2$ of four liver lobes of mice infected with either PfSPZ-WT or PfSPZ-LARC2. Statistical analysis was carried out using unpaired t-test. $P > 0.05$ is taken as ns. (C) Comparison of the size of LS parasites (based on area at the parasite's largest circumference) between PfSPZ-WT and PfSPZ-LARC2 at 6- and 7-days post infection. There is no significant difference in LS size between PfSPZ-WT and PfSPZ-LARC2. Data is represented as mean ± SD. Each datapoint refers to the mean size of at least 25 parasites for each timepoint. Statistical analysis was carried out using two-way ANOVA using Tukey's multiple comparison test. $P > 0.05$ is taken as ns. Liver stage development and differentiation was compared between PfSPZ-WT and PfSPZ-LARC2 at 7 days post infection using antibodies against; (D) the PPM protein CSP (green), the PVM protein Exp1 (red), and, mature LS stage merozoite markers (E) MSP1 and (F) mTIP on day 7 post infections. DNA is stained with DAPI. Scale bar is 20 μm. PfSPZ-LARC2 late LS schizonts display aberrant distribution of CSP in late LS schizonts, in contrast to in PFSPZ-WT, where CSP is localized to the PPM and delineates cytomeres. No PfSPZ-LARC2 LS merozoite formation was observed. MSP-1 expression was very weak and there was a complete lack of expression of mTIP. In contrast, PfSPZ-WT LS expressed both MSP1 and mTIP and these proteins delineated LS stage merozoites. Source data are available online for this figure.

LARC2-infected FRG NOD huHep mice on day 7 post infection. Notwithstanding, 6 weeks of continuous in vitro culture of the blood harvested from PfSPZ-LARC2 infected FRG NOD huHep mice did not show growth of any blood stage parasites. In contrast, blood stage parasites were robustly observed within 1 to 3 days of in vitro blood culture after transition from mice injected with one million aseptic, purified, cryopreserved PfSPZ-WT. It is thus possible that the PfLARC2 18S rRNA signal originated from the parasite nucleic acid released into circulation by dying and disintegrating PfLARC2-infected hepatocytes. Nonetheless, the absence of any viable parasites after prolonged culture affirms complete attenuation of the PfLARC2 parasite.

In summary, we have generated the Pf malaria vaccine candidate PfSPZ-LARC2 and shown preclinical evidence that it can be manufactured and vialed. PfSPZ-LARC2 undergoes late LS developmental arrest and fails to generate infectious LS merozoites in the humanized mouse model. We have also provided conclusive evidence using the rodent malaria model that PySPZ-LARC2 is a potent immunogen and confers broad stage-transcending and species-transcending protection. Any whole-cell investigational vaccine product for use in humans must be devoid of extraneous DNA. We confirmed by both whole genome sequencing and extensive PCR genotyping that PfLARC2 clone F7 lacks extraneous DNA and is thus suitable for human use. Clinical development of PfSPZ-LARC2 Vaccine will be initiated with dose escalation safety studies in malaria-naive adults followed by one-dose to three-dose vaccination regimens that will evaluate protection against CHMI with the Pf 7G8 strain (Silva et al, 2022), which is highly genetically divergent from the NF54 strain used to create PfLARC2 (Silva et al, 2022). In concert, an age de-escalation safety trial in malaria-exposed Africans is being planned.

# Methods

## Study approval

This study was carried out in accordance with the recommendations of the NIH Office of Laboratory Animal Welfare standards (OLAW welfare assurance #D16-00119). Mice were maintained and bred under specific pathogen-free conditions at the Center for Global Infectious Disease Research, Seattle Children's Research Institute. The protocols were approved by the Center for Infectious Disease Research Institutional Animal Care and Use Committee (IACUC) under protocol 00480 (FRG NOD huHep mice) and 00505 (other rodent strains).

## Experimental animals

Six- to eight-week-old female Swiss Webster (SW) mice were purchased from Envigo and used for Py parasite life cycle maintenance and production of transgenic parasites. Six- to eight-week-old female BALB/cJ and BALB/cByJ mice were purchased from Jackson Laboratories and used for assessments of parasite infectivity, indirect immunofluorescence assays (IFA), as well as the attenuation and ability of Py LARC2 parasite to act as an experimental vaccine. For studies of Pf LARC2, FRG NOD huHep mice (female and >4 months of age) were purchased from Yecuris, Inc. Repopulation of human hepatocytes was confirmed by measuring human serum albumin levels, and only animals with human serum albumin levels >4 mg/mL were used, corresponding to >70% human hepatocyte repopulation. FRG NOD huHep mice were maintained on drinking water containing 3% Dextrose and were cycled on 8 mg/L NTBC once a month for 4 days to maintain hepatocyte chimerism. Py 17XNL wildtype and transgenic parasites were cycled between SW mice and *A. stephensi* mosquitoes for the purposes of sporozoite production. Infected mosquitoes were maintained on sugar water at 24 °C and 70% humidity. The assignment of which animals would be infected with wild type vs transgenic parasites was performed randomly. Further downstream processing, including harvesting blood and tissue samples were not performed in a blinded fashion.

## Creation of a P. yoelii LARC2

All oligonucleotide primers used for the creation and analysis of Py LARC2 are detailed in Appendix Table S1. For generation of PyLARC2 (Py *mei2⁻/linup⁻*), a marker free clone of Py *mei2⁻* (Vaughan et al, 2018) was used for deletion of Py LINUP using CRISPR/Cas9 methodology. pYC_LINUP was transfected into the blood stage schizonts of Py *mei2⁻*. After transfection and intravenous injection into SW mice, pyrimethamine was used for the positive selection and downstream cloning of recombinant parasites using standard techniques. Gene knockout was confirmed by PCR using methodology we have used on multiple occasions (see (Vaughan et al, 2018) for a recent example). This led to the creation of the PyLARC2 and two separate knockout clones from

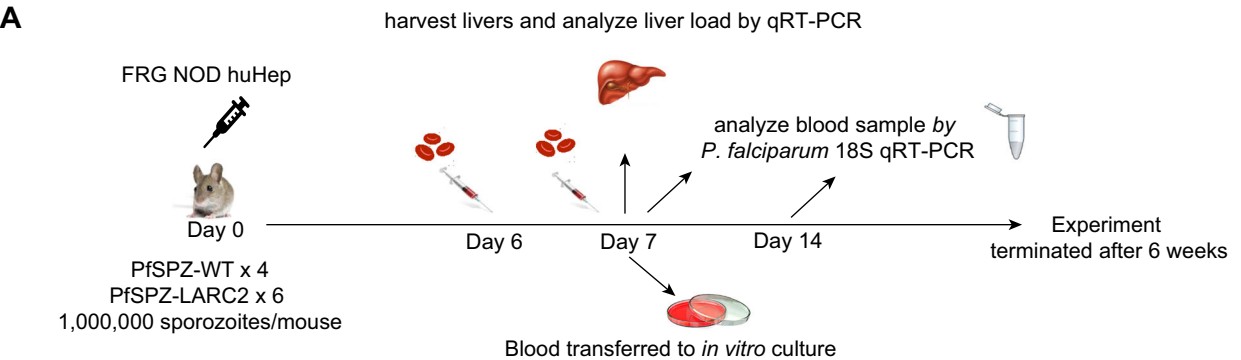

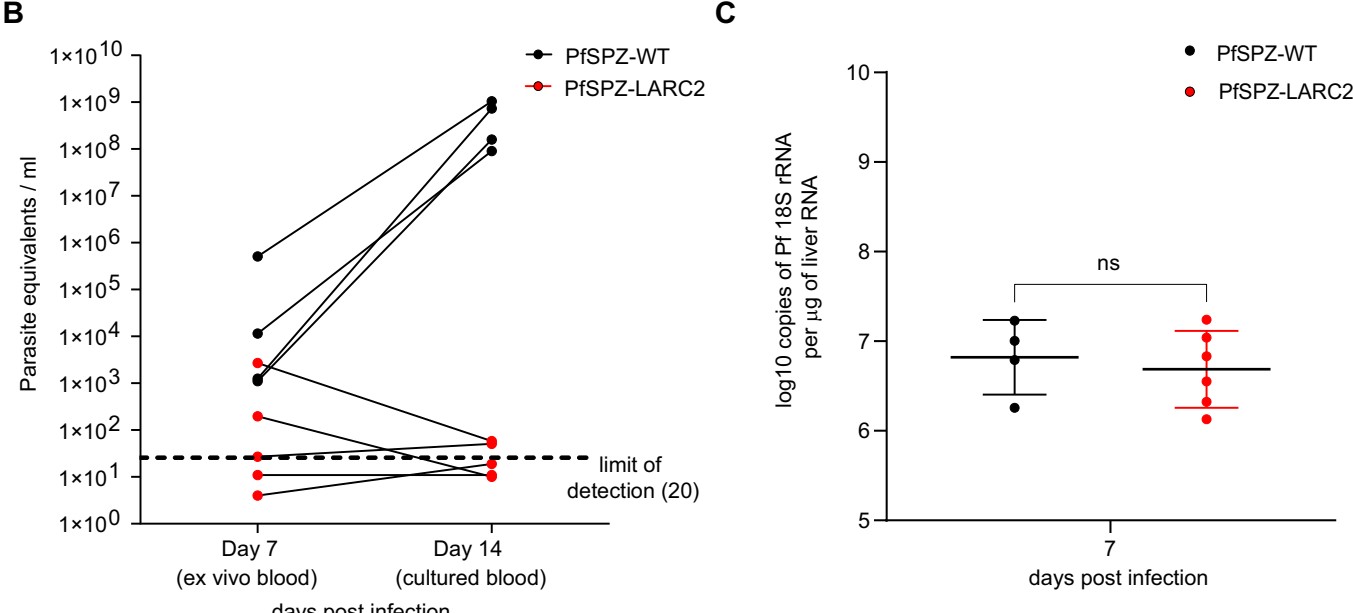

**Figure 7. PfSPZ-LARC2 parasites do not transition to viable blood stage infection.**

(A) The schematic depicts the experimental design. To test blood stage breakthrough of *P*. PfLARC2, 1 million of aseptic, cryopreserved PfSPZ-WT or PfSPZ-LARC2 sporozoites were injected i.v. into four or six FRG NOD huHep mice per group, respectively. On days 6 and 7, 400 µl of 70% human RBCs were injected i.v. to enable transition of LS parasites to blood stage infection. Four hours after human RBC repopulation on day 7, mice were euthanized, blood was collected by cardiac puncture and 50 µl of blood from each mouse was used for qRT-PCR analysis to detect parasite 18S rRNA in the blood. Liver samples were harvested for quantification of parasite liver stage burden by qRT-PCR. The remaining blood was transferred to in vitro asexual parasite culture. Fresh media was replaced daily, and cultures were analyzed every 2 to 3 days by thick smear for presence of parasites for up to 6 weeks. (B) On day 7, parasite blood stage densities for mice infected with PfSPZ-WT ranged between $10^3$–$10^6$ parasite equivalents/ml by Pf 18S qRT-PCR. Parasite densities increased to ~$10^8$–$10^9$ parasite equivalents/ml after 7 days of in vitro culture (14 days after PfSPZ-WT sporozoite infection). Blood stage parasites were detected by Giemsa-stained smears in all four mice infected with PfSPZ-WT within 1 to 3 days of transition to in vitro culture (see Table 6; Appendix Table S2). In contrast, no parasites were detected by Giemsa-stained thin smears in all six mice infected with PfSPZ-LARC2. (C) Livers of four PfSPZ-WT infected mice and six PfSPZ-LARC2 infected mice were harvested on day 7 post infection for analysis of parasite liver stage burden by Pf 18S qRT-PCR. Parasite liver stage load is depicted as log10 copies of Pf 18S rRNA per µg of total liver RNA. Data is presented as mean ± SD. Statistical analysis was carried out using unpaired t-test. $P > 0.05$ is taken as ns. Source data are available online for this figure.

two independent transfections were initially phenotypically analyzed throughout the life cycle.

#### Production of aseptic, purified, cryopreserved PfSPZ-LARC2 and vialing of cryopreserved *P. falciparum* LARC2

PfLARC2 for clone F7 was optimized for in vitro production of gametocytes and robust yields of PfSPZ-LARC2, generated in aseptic mosquitoes (Hoffman et al, 2010). Production and characterization of aseptic, purified PfSPZ-LARC2 was as described previously (Seder

et al, 2013; Hoffman et al, 2010; Epstein et al, 2011). PfSPZ-LARC2 was vialed and cryopreserved in liquid nitrogen vapor phase (below −150 °C) at a concentration of $2 \times 10^5$ PfSPZ/vial in 20 µL volumes.

#### PyLARC2 sporozoite isolation and evaluation of LS attenuation

PyLARC2 sporozoites were isolated from the salivary glands of infected *A. stephensi* mosquitoes between 14 and 18 days after the infectious blood meal and injected intravenously into the tail vein of recipient mice. For assessment of Py LARC2 attenuation,

**Table 6. PfSPZ-LARC2 liver stages do not transition to productive blood stage infection after sporozoite infection in FRG huHep.**

| Mice | Blood stage parasites by thin Giemsa smear (days post transition) |
|---|---|
| PfSPZ_1 | Positive (day 2 after transition) |
| PfSPZ_2 | Positive (day 3 after transition) |
| PfSPZ_3 | Positive (day 2 after transition) |
| PfSPZ_4 | Positive (day 1 after transition) |
| PfSPZ-LARC2_1 | Negative upto 6 weeks |
| PfSPZ-LARC2_2 | Negative upto 6 weeks |
| PfSPZ-LARC2_3 | Negative upto 6 weeks |
| PfSPZ-LARC2_4 | Negative upto 6 weeks |
| PfSPZ-LARC2_5 | Negative upto 6 weeks |
| PfSPZ-LARC2_6 | Negative upto 6 weeks |

250,000 sporozoites were injected intravenously into highly susceptible BALB/cByJ mice. Liver stage-to-blood stage transition (blood stage patency) was assessed by Giemsa-stained thin blood smear starting at day 3 after inoculation and ending at day 14, at which time, a negative smear was attributed to complete attenuation.

### Immunization and challenge

BALB/c, C57Bl/6 or Swiss Webster mice were immunized intravenously with two or three doses of PyLARC2 sporozoites. For some experiments, Balb/c mice were immunized through intramuscular (i.m.) route. The immunization doses/regimens are described in the figure legends. Mice immunized with uninfected mosquito salivary gland extract were used as mock controls. The immunized mice were subsequently challenged with either homologous wildtype *P. yoelii*[GFP-luciferase] (Miller et al, 2013) or heterologous *P. berghei* ANKA sporozoites at different time duration after the last immunization as explained in the figure legends. Protection was assessed by examining Giemsa-stained thin blood smear, made from mouse tail snip bleeding, starting from day 3 to 14 after challenge. Mice were considered sterile protected if blood stage parasites were absent while examining at least 30 microscopic fields per smear. For examining protection specifically against lethal *Plasmodium* blood stage infection, PyLARC2 immunized animals were directly challenged by i.v. injection of 10,000 Py XL infected RBCs (iRBCs). The protection was assessed by counting the parasitemia between day 1 to 14 after blood stage challenge. For the assessment of antibody reactivity to LS and blood stage parasites, blood samples from PyLARC2 or mock immunized animals were collected in heparin tubes via retro-orbital bleeding at 2 weeks after the boost. The plasma was harvested by centrifuging blood at $4500 \times g$ for 10 min at 4 °C and then stored at $-20$ °C until further use (Vigdorovich et al, 2023).

### Cell isolation and flow cytometry

Cells were harvested as described before (Patel et al, 2017, 2019). Livers were perfused by injecting 10 ml sterile 1x PBS via hepatic portal vein to achieve uniform blanching and remove the non-resident lymphocytes. After perfusion livers and spleen were collected in 1% FACS buffer (1% fetal bovine serum in 1x PBS).

Single cell suspension was prepared by mechanically disrupting the organs through 70 µm nylon cell strainers using 10 ml syringe plunger. Liver cells were resuspended in sterile PBS with 35% Percoll (GF Healthcare) and centrifuged at $800 \times g$ for 20 min at RT. The red blood cells were lysed by incubating liver and splenic cells with ACK lysis solution for 4 min. Final hepatic non-parenchymal cells and splenocytes were resuspended in 1% FACS buffer, further counted and used for flow cytometric analysis. Cells were incubated with mouse Fc block (Biolegend #156604) for 10 min at 4 °C and stained with cell surface antibodies at 1:100 dilution specific for CD3-BV605 (clone 17A2), CD19-PECy5.5 (clone eBio1D3), CD8-BV785 (clone 53-6.7), CD4-PE-Dazzle594 (clone 100455), CD44-BV510 (clone IM7), CD62L-APC-R700 (clone MEL-14), CXCR6-BV421 (clone SA051D1), and CD69-PECy7 (clone H1.2F3) for 20 min at 4 °C. Cells were acquired in BD FACS LSRII flow cytometer and data were analyzed with FlowJo v10. software (BD).

### In vitro culturing of P. falciparum parasite lines

Pf strain NF54 was used as the parent wildtype strain for all transgenesis and as the control strain in experiments. Pf NF54, Pf *mei2*[-] and Pf parasite lines were cultured in custom-made RPMI 1640 media containing hypoxanthine, sodium bicarbonate, and 4-(2-hydroxyethyl)-1-piperazineethanesulfonic acid (Invitrogen, Life Technologies, Grand Island, NY) supplemented with 5% human serum (Valley Biomedical), 5% AlbuMAX (Invitrogen, Life Technologies, Grand Island, NY) and gentamycin (Invitrogen), in presence of O[+] blood (Valley Biomedical) at 3–5% hematocrit, in an atmosphere of 5% $CO_2$, 5% $O_2$, and 90% $N_2$.

### Creation of the P. falciparum LARC2 parasite

Marker-free Pf *mei2*[-] clone F3 (Goswami et al, 2020) was used for deletion of Pf LINUP using CRISPR/Cas9 methodology. 5% sorbitol synchronized ring stage Pf *mei2*[-] parasites at 5–8% parasitemia were electroporated with 100 µg of pFC-LINUP plasmid at 0.31 kV and 950 µF using a BioRad Gene Pulser (BioRad, Hercules, CA). Positive selection with 8 nM WR99210 (WR; Jacobus Pharmaceuticals, Princeton, NJ) was administered 24 h after the transfection and kept for 5 days. Once drug-resistant parasites were visible on thin blood smears (day 15), parasites were put through two rounds of negative selection pressure with 1 µM 5-fluorocytosine (Cayman Chemical) for 7 days each and cloned by limiting dilution. These parasites were screened by PCR using gene-specific primers (Appendix Table S1) and whole genome sequencing to confirm deletion of both Pf Mei2 and LINUP and absence of plasmid DNA.

### P. falciparum parasite cloning by limiting dilution

PfLARC2 parasites were cloned out of the mixed parasite pool by limiting dilution in 96-well flat-bottomed plates. Cultures were serially diluted and then plated at a density of 0.3 parasites per well in a 100 µl volume at 2% hematocrit. Cultures were fed once a week and fresh blood was added every 3–4 days at 0.3% hematocrit. The wells were screened for presence of parasites using primers to 18S RNA on day (Appendix Table S1) [18SF (5' AACCTGGTTGATCC AGTAGTCATATG 3') and 18SR (5' CCAAAAATTGGCCTTGC ATTGTTAT 3')]. Positive wells were expanded and screened for recombination using the PCR strategies used above. PfLARC2 clone F7 and B7 were used for further phenotypic analysis.

### Gametocyte culture and mosquito infections

Pf NF54-WT and PfLARC2 cultures were grown in gametocyte media containing RPMI 1640 media containing hypoxanthine, sodium bicarbonate, and 4-(2-hydroxyethyl)-1-piperazineethane-sulfonic acid (Invitrogen, Life Technologies, Grand Island, NY) supplemented with 10% human serum (Valley Biomedical) in presence of O+ blood (Valley Biomedical) at 4% hematocrit, in an atmosphere of 5% $CO_2$, 5% $O_2$, and 90% $N_2$. Gametocyte cultures were initiated at 4% hematocrit and 0.8–1% parasitemia (mixed stages) and maintained for up to 15 days with daily medium changes. On days 15–17 post set up, stage V gametocytemia was evaluated by Giemsa-stained thin blood smears. Mature gametocyte cultures were spun down at $800 \times g$ for 2 mins at 37 °C to pellet infected red blood cells. A volume of pre-warmed 100% human serum equal to the packed red blood cell volume was added to resuspend the red blood cell pellet. The resuspended pellet was diluted to a gametocytemia of 0.4–0.5% with pre-warmed feeding media containing 50% human serum and 50% human red blood cell and immediately fed to non-blood fed adult female mosquitoes 3 to 7 days after hatching by membrane feeding assay. Mosquitoes were allowed to feed through Parafilm for up to 20 min. Following blood feeding, mosquitoes were maintained for up to 19 days at 27 °C, 75% humidity, and provided with 8% dextrose solution in PABA water. Oocyst prevalence was checked on days 7–9 post feed by microdissecting approximately 12 midguts per cage. Sporozoite numbers were detected by dissecting and grinding salivary glands in Schneider's Insect Medium (Sigma) on days 14–18 post feed and at least 50 mosquitoes were dissected per strain.

### Phenotypic analysis of PfLARC2 parasites in FRG NOD huHep mice

Aseptic, purified, cryopreserved PfNF54-WT (PfSPZ-WT) and PfLARC2 (PfSPZ-LARC2) sporozoites were used to infection of FRG NOD huHep mice. For evaluation of LS development of PfLARC2, three million PfSPZ-WT and three million PfSPZ-LARC2 sporozoites were injected intravenously (retro-orbital) into two FRG NOD huHep mice per group. Livers were harvested on days 6 and 7 and used for IFA. To evaluate for blood stage transition of PfSPZ-LARC2 in FRG NOD huHep mice, one million PfSPZ-WT and one million PfSPZ-LARC2 sporozoites were injected intravenously into four and six FRG NOD huHep mice per group, respectively. On days 6 and 7, 400 μl of 70% RBCs were injected intravenously to enable transition of LS parasites to blood. Four hours after human RBC repopulation on day 7, mice were euthanized, blood was collected by cardiac puncture and 50 μl of blood from each mouse was used for qRT-PCR analysis to detect parasite 18S RNA and livers were harvested for RNA extraction to quantify parasite liver load. Blood was washed three times in asexual media, a volume of human RBCs equal to the packed RBC volume was added and blood was transferred to in vitro culture. Fresh media was replaced daily, and cultures were analyzed every 2 to 3 days by thick smear for presence of parasites for up to 6 weeks. Samples from in vitro culture were analyzed for presence of 18S rRNA by qRT-PCR after 7 days.

### Plasmodium 18S rRNA qRT-PCR quantification of parasite load

For quantification of *Plasmodium* 18S rRNA from mouse blood, 50 μL of whole blood was added to 2 mL of NucliSENS lysis buffer (bioMérieux, Marcy-l'Étoile, France) and frozen immediately at −80 °C. One mL of lysate was subsequently processed using the Abbott m2000sp using mSample RNA preparation kit (Abbott, Niles, IL). The eluate was tested by qRT-PCR for the pan-*Plasmodium* 18 S rRNA target. The qRT-PCR reaction was performed using 35 μL SensiFAST™ Probe Lo-ROX One-Step Kit (Bioline, Taunton, MA) and 15 μL of extracted eluate. Plasmodium 18 S rRNA primers/probes (LCG BioSearch Technologies, Novato, CA) were as follows: Forward primer PanDDT1043F19 (0.2 μM): 5'-AAAGTTAAGGGAGTGAAGA-3'; Reverse primer PanDDT1197R22 (0.2 μM): 5'-AAGACTTTGATTTCTCATAA GG-3'; Probe (0.1 μM): 5'-[CAL Fluor Orange 560]-ACCGTCG TAATCTTAACCATAAACTA[T(Black Hole Quencher-1)]GCCG ACTAG-3'[Spacer C3]). Cycling conditions were RT (10 min) at 48 °C, denaturation (2 min) at 95 °C and 45 PCR cycles of 95 °C (5 s) and 50 °C (35 s).

### Immunofluorescence assay (IFA)

For IFA for LS, Livers were perfused with 1×PBS, fixed in 4% v/v paraformaldehyde (PFA) in 1×PBS and lobes were cut into 50 μm sections using a Vibratome apparatus (Ted Pella, Redding, CA). For IFA, sections were permeabilized in 1×TBS containing 3% v/v $H_2O_2$ and 0.25% v/v Triton X-100 for 30 min at room temperature. Sections were then blocked in 1×TBS containing 5% v/v dried milk (TBS-M) for at least 1 h and incubated with primary antibody in TBS-M at 4 °C overnight. After washing in 1×TBS, fluorescent secondary antibodies were added in TBS-M for 2 h at room temperature in a similar manner as above. After further washing, the section was incubated in 0.06% w/v $KMnO_4$ for 2 min to quench background fluorescence. Sections were then washed with 1×TBS and stained with 1 μg/ml 4',6-diamidino-2-phenylindole (DAPI) in 1×TBS for 5–10 min at room temperature to visualize DNA and mounted with FluoroGuard anti-fade reagent (Bio-Rad, Hercules, CA).

For assessment of reactivity of LARC2-immunized sera to antigens expressed on PyWT LS, BALB cBy/J mice were infected with 250,000 PyWT sporozoites and livers were harvested at 24, 32, and 48 hpi and processed for IFA as written above, with Py LARC2-immunized anti-sera used as primary antibody.

Other primary antibodies used included Pf CSP clone 2A10 (1:500, BEI Resources, Cat # BEI-MRA-183A, mouse monoclonal), Py ACP clone 40001 (1:500, Stefan Kappe, mouse monoclonal) (Mikolajczak et al, 2015), *P. vivax* mHSP60 (1:1000, Stefan Kappe, rabbit polyclonal) (Mikolajczak et al, 2015), Pf MSP-1 clone 12.1 (1:500, European Malaria Reagent Repository, http://www.malariaresearch.eu, mouse monoclonal) (Conway et al, 1991), Py mTIP (1:500, Stefan Kappe, rabbit polyclonal) (Bergman et al, 2003), Pf Exp1 (1:500, Stefan Kappe, rabbit polyclonal) (Goswami et al, 2022a) and mouse anti-sera (1:100). Secondary antibodies include donkey anti-mouse 488 (Invitrogen, Cat # A21202; RRID: AB_141607), donkey anti-mouse 594 (Invitrogen, Cat # A21203; RRID: AB_141633), donkey anti-rabbit 488 (Invitrogen, Cat # A21206; RRID: AB_2535792) and donkey anti-rabbit 594 (Invitrogen, Cat # A21207; RRID: AB_141637). All secondary antibodies were used at a dilution of 1:1000.

To analyze the reactivity of PyLARC2-immunized anti-sera against PyWT blood stage antigens, thin blood smears of PyWT parasites were fixed in 100% ice-cold methanol processed for 3 min followed by two washes in 1X PBS at room temperature. Slides were marked with blocked in 2% bovine serum albumin (BSA) for 1 h at room temperature, followed by incubation with PyLARC2 anti-sera

**The paper explained**

**Problem**

*Plasmodium falciparum* malaria parasites continue to take an enormous toll on global human health. Therefore, there is an urgent need for a highly efficacious vaccine capable of conferring sterilizing immunity and thereby preventing malaria parasite infection. Despite decades of development, the currently approved subunit malaria vaccines only provide partial protection against malaria, necessitating further advancements in malaria vaccine development.

**Results**

In this study, we engineered an attenuated *P. falciparum* strain called PfLARC2 as well as an equivalent LARC2 in a mouse malaria model. This whole parasite vaccine is engineered to lack two genes essential for parasite transition from asymptomatic liver stage infection to symptomatic blood stage infection. In the mouse model, LARC2 immunizations are safe and stimulate robust humoral and cellular immune responses. This confers sterilizing immunity against infection. In liver humanized mouse infections, PfSPZ-LARC2 is replication-competent but completely attenuated, and potentially expresses thousands of liver stage parasite antigens. A vialed version of PfLARC2, referred to as PfSPZ-LARC2, has been generated and will be tested in clinical trials.

**Impact**

PfSPZ-LARC2 is safe and potent in animal models and is now considered the leading whole cell malaria vaccine candidate that aims to prevent infection. Human safety and efficacy will be evaluated in clinical studies and if favorable, this vaccine can contribute to malaria elimination and eradication.

(1:100) and Py MSP1 (1:200) in 2% BSA overnight at 4 °C. Slides were washed three times in 1X PBS, 8 min per wash, followed by incubation with secondary antibodies (1:1000) and DAPI (1 mg/mL) in 2% BSA for 1 h. Slides were washed three times in 1X PBS, 8 min per wash and mounted in ProLong™Gold Antifade Mountant (Molecular Probes, CA) and a coverslip.

### Microscopy

All images were acquired using the Stellaris 8 confocal microscope (Leica Microsystems) with 63x water objective and processed using the Lightning software (Leica Microsystems). For quantification of parasite LS size, the parasite was assumed to be elliptical in shape and therefore area was calculated from its longest (a) and shortest (b) circumferential diameter ($\pi ab$).

### Quantification and statistical analysis

Where appropriate, quantification was represented by the mean ± standard deviation. Calculations and statistical tests indicated in the figure legends were performed using GraphPad Prism Software. The statistical tests used was two-way ANOVA. $*P < 0.05$, $**P < 0.01$, $***P < 0.001$. A $P$ value $> 0.05$ was considered not significant.

### Graphics

The synopsis figure was designed using BioRender.com.

For more information please refer to https://www.seattlechildrens.org/research/centers-programs/global-infectious-disease-research/research-areas-and-labs/kappe-lab/.

## Data availability

This study includes no data deposited in external repositories.

## Peer review information

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

## Acknowledgements

We thank the insectary team at the Center for Global Infectious Disease Research for providing *Anopheles stephensi* mosquitoes. This work was funded by the NIH-NIAID grants, UO1-AI155335 and SBIR phase I (R43AI149927) and II (2R44AI149927) grants.

## Author contributions

**Debashree Goswami**: Conceptualization; Resources; Data curation; Software; Formal analysis; Supervision; Funding acquisition; Validation; Investigation; Visualization; Methodology; Writing—original draft; Project administration; Writing—review and editing; Conceived the work and designed experiments. Carried out the research and collected and analyzed the data. Wrote the manuscript. **Hardik Patel**: Conceptualization; Resources; Data curation; Software; Formal analysis; Supervision; Funding acquisition; Validation; Investigation; Visualization; Methodology; Writing—original draft; Project administration; Writing—review and editing; Conceived the work and designed experiments. Carried out the research and collected and analyzed the data. Wrote the manuscript. **William Betz**: Resources; Data curation; Investigation; Visualization; Methodology; Carried out the research and collected and analyzed the data. **Janna Armstrong**: Data curation; Formal analysis; Validation; Investigation; Methodology; Carried out the research and collected and analyzed the data. **Nelly Camargo**: Data curation; Methodology; Carried out the research and collected and analyzed the data. **Asha Patil**: Resources; Data curation; Formal analysis; Validation; Investigation; Visualization; Methodology; Produced cryopreserved PfPSZ-WT and, PfSPZ-LARC2 for experiments with FRG NOD huHep and cryopreserved PyLARC2 for rodent immunization/challenge studies. **Sumana Chakravarty**: Data curation; Validation; Investigation; Methodology; produced cryopreserved PfPSZ-WT and, PfSPZ-LARC2 for experiments with FRG NOD huHep and cryopreserved PyLARC2 for rodent immunization/challenge studies. **Sean C Murphy**: Data curation; Formal analysis; Validation; Investigation; Methodology; Analyzed the qRT-PCR data. **B Kim Lee Sim**: Data curation; Supervision; Validation; Investigation; Methodology; Writing—review and editing; Produced cryopreserved PfPSZ-WT and, PfSPZ-LARC2 for experiments with FRG NOD huHep and cryopreserved PyLARC2 for rodent immunization/challenge studies. Edited the manuscript. **Ashley M Vaughan**: Conceptualization; Supervision; Funding acquisition; Validation; Methodology; Project administration; Writing—review and editing; Conceived the work and designed experiments and edited the manuscript. **Stephen L Hoffman**: Conceptualization; Data curation; Formal analysis; Supervision; Funding acquisition; Investigation; Methodology; Writing—original draft; Writing—review and editing; Produced cryopreserved PfPSZ-WT and, PfSPZ-LARC2 for experiments with FRG NOD huHep and cryopreserved PyLARC2 for rodent immunization/challenge studies. Edited the manuscript. **Stefan HI Kappe**: Conceptualization; Resources; Supervision; Funding acquisition; Writing—original draft; Writing—review and editing; Conceived the work and designed experiments, wrote and edited the manuscript.

## Disclosure and competing interests statement

Stefan Kappe, Ph.D., Ashley Vaughan, Ph.D., and Debashree Goswami, Ph.D. are members of the Seattle Children's Hospital workforce, are inventors of GAP patents licensed to Sanaria and have revenue distribution rights arising from patents rights licensed to Sanaria.

# Expanded View Figures

**Figure EV1. PyLARC2 immunization elicits long-lived CD8 + T cell-based immunity, especially liver resident CD8 + T cells.**

(**A**) Experimental schematics for (**B–E**). Nine C57BL/6 mice were immunized three times with 50,000 PyLARC2 sporozoites 1 month apart. Five mock immunized mice received mosquito debris from uninfected salivary glands as a control. Mice were euthanized at 2 months after the last immunization and the livers and spleens were collected for T cell analysis via flow cytometry. (**B**) (i) frequencies and (ii) the total number of antigen experienced CD8$^+$ memory T cells (CD44$^{hi}$) in the livers of mock or PyLARC2 immunized mice. (**C**) (i) frequencies, (ii) total number, and (iii) representative flow layout of CD69$^{hi}$CXCR6$^{hi}$liver-resident memory CD8$^+$ T cells (T$_{RM}$) (gated on CD44$^{hi}$CD8α$^+$ singlet events). (**D**) Frequencies and a representative flow layout of CD11a$^{hi}$CD44$^{hi}$ CD8$^+$ T cells in the liver. (**E**) (i) frequencies, (ii) total number, and (iii) representative flow layout of CD44$^{hi}$CD62L$^{lo}$ CD8$^+$ effector memory T cells (T$_{EM}$) in the spleen. Data is shown as mean ± SD. Statistical analysis was done using non-parametric Mann–Whitney test. *P* values are indicated in the figure. *P* > 0.05 is taken as not significant.

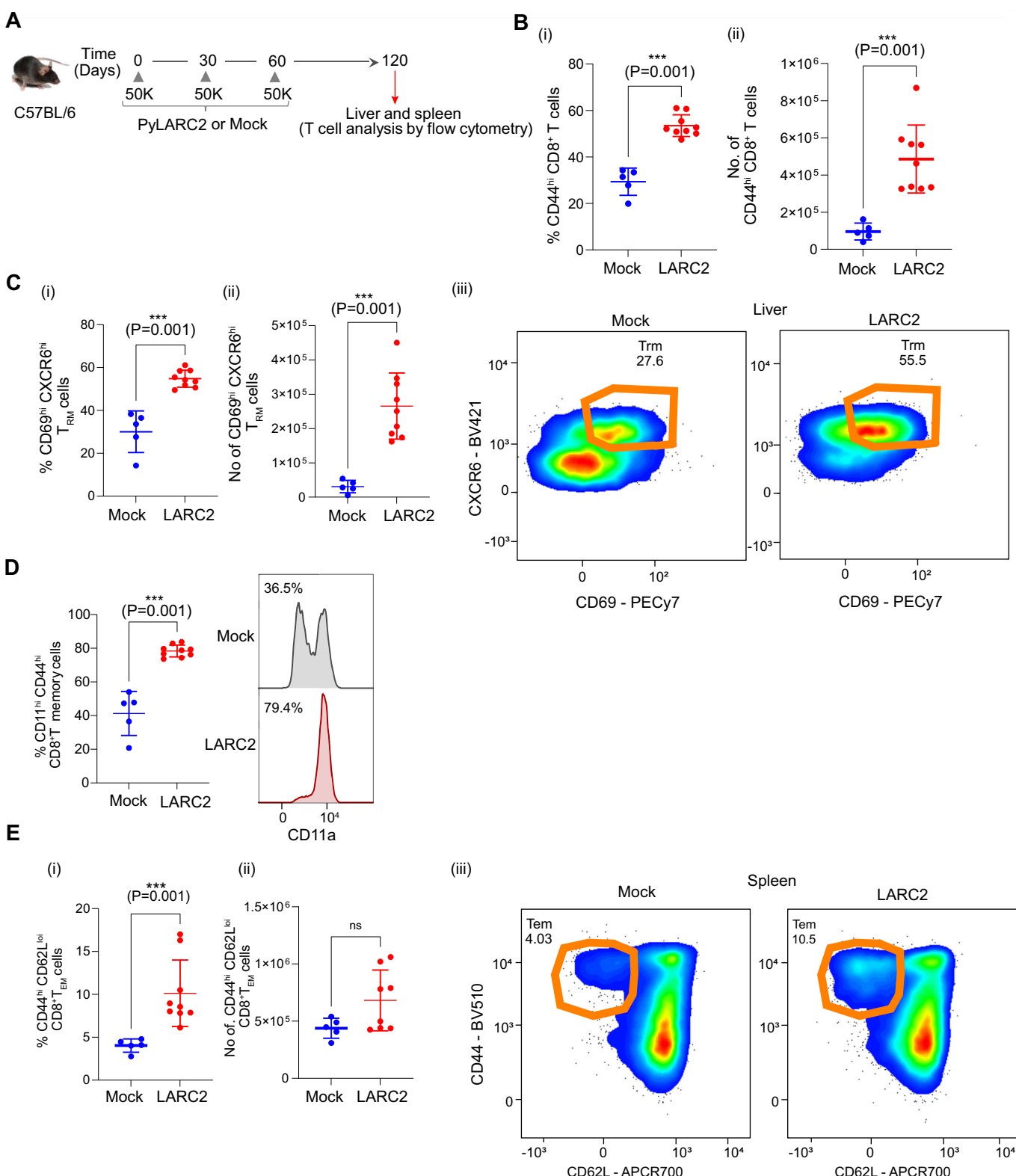

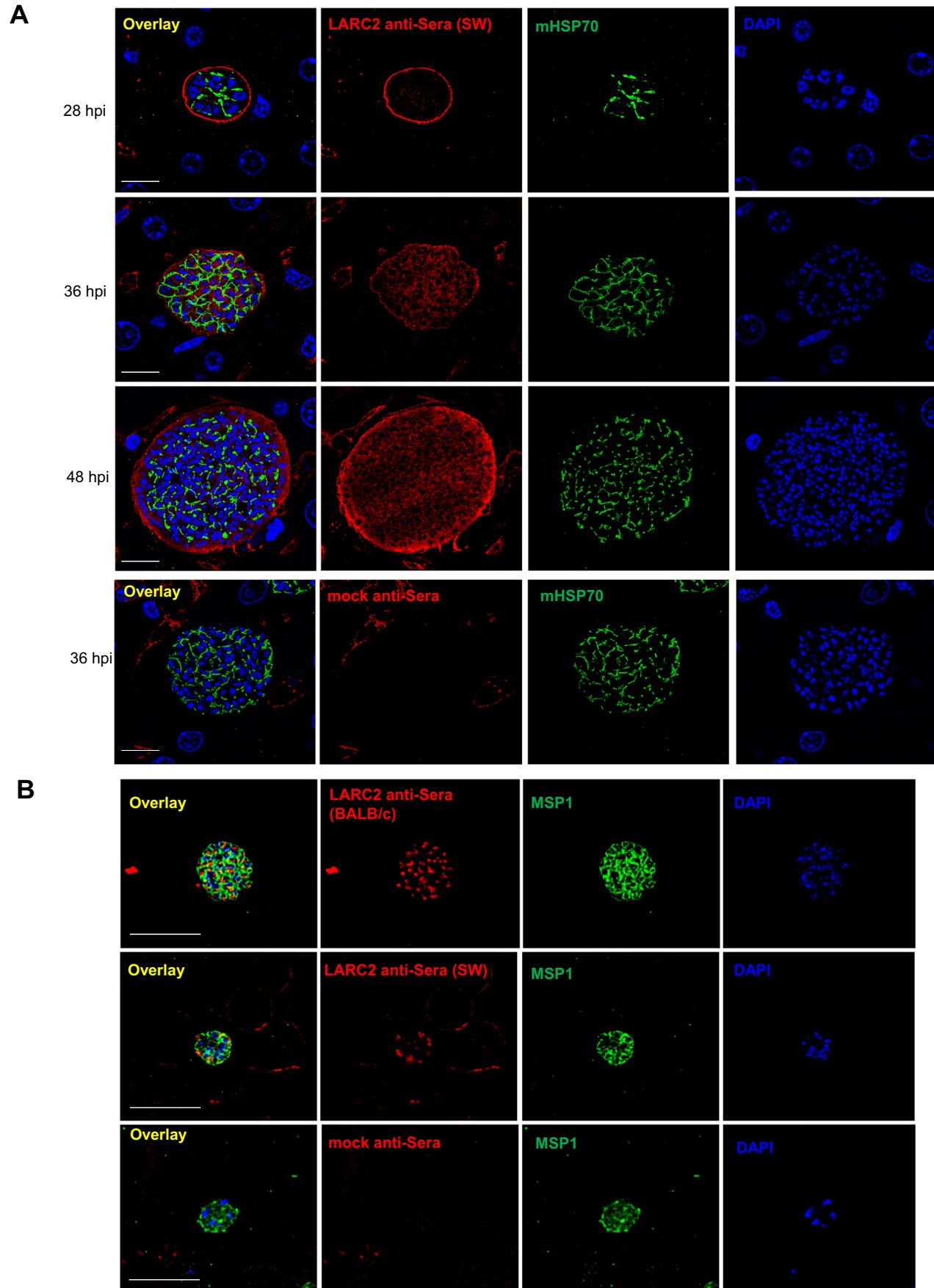

◄  **Figure EV2.   PyLARC2 immunization of elicits humoral responses to liver and blood stages.**

(A) Plasma was collected from six PyLARC2 immunized SW mice and pooled. The pooled plasma was then diluted 1:100 and used to evaluate reactivity to antigens at different timepoints of PyWT LS development using IFA. Sera is shown in red, anti-mHSP70 antibody is used as control (green) and DNA is stained with DAPI (blue). Scale bar is 20 µm. At early timepoints of LS development, the immune sera from PyLARC2 immunized mice reacted with antigens expressed on the PVM or PPM. In more mature LS, both cytoplasmic and membrane reactivity was observed, indicating that the serum antibodies recognized antigens located in both the cytoplasm and the membranes of the parasite. No reactivity was observed in plasma harvested from mock-immunized mice, indicating that the staining was specific to the PyLARC2 immunized mice. (B) Plasma was collected from six PyLARC2 immunized SW mice or BALB/cJ mice. Pooled plasma for each mouse strain was diluted 1:100 and used to evaluate reactivity to antigens expressed in PyWT blood stages. Sera is shown in red, anti-MSP1antibody is used as control (green) and DNA is stained with DAPI (blue). Scale bar is 10 µm. Immune sera from both SW and BALB/cJ mice reacted with antigens expressed in the cytoplasm of blood stages. No reactivity was observed in plasma harvested from mock-immunized mice, indicating that the staining was specific to immune sera from PyLARC2 immunized mice.

   

**A**

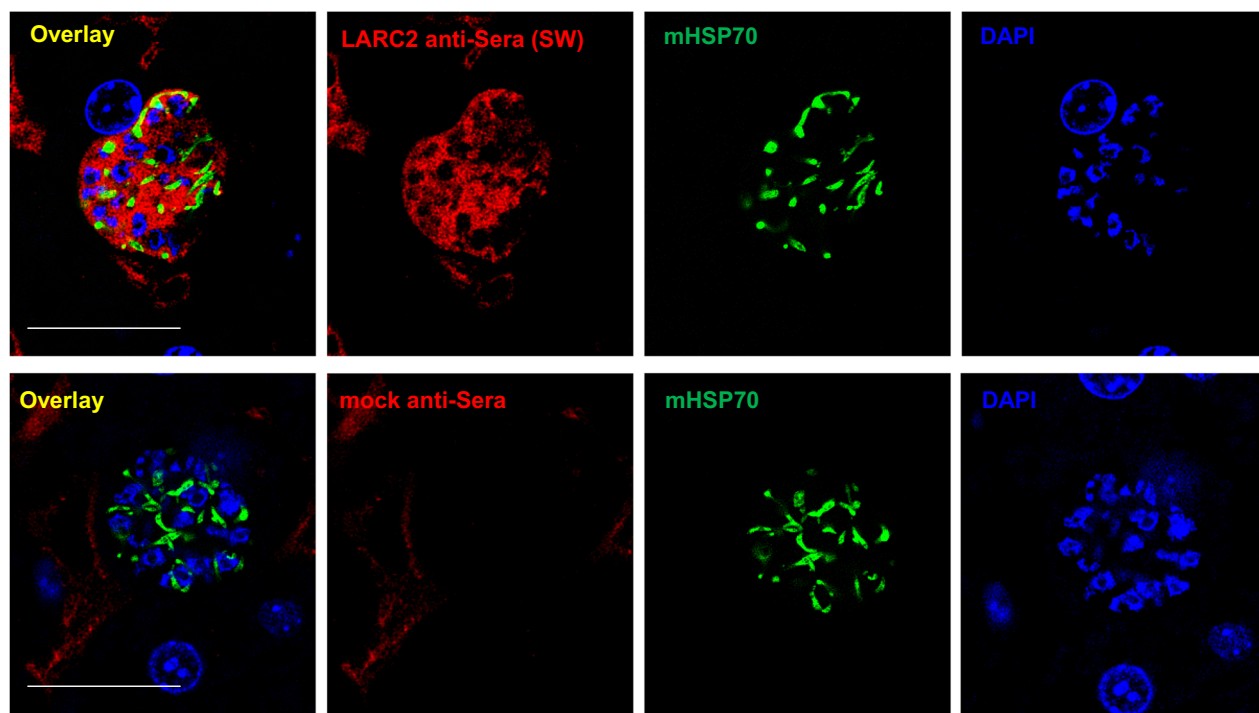

**B**

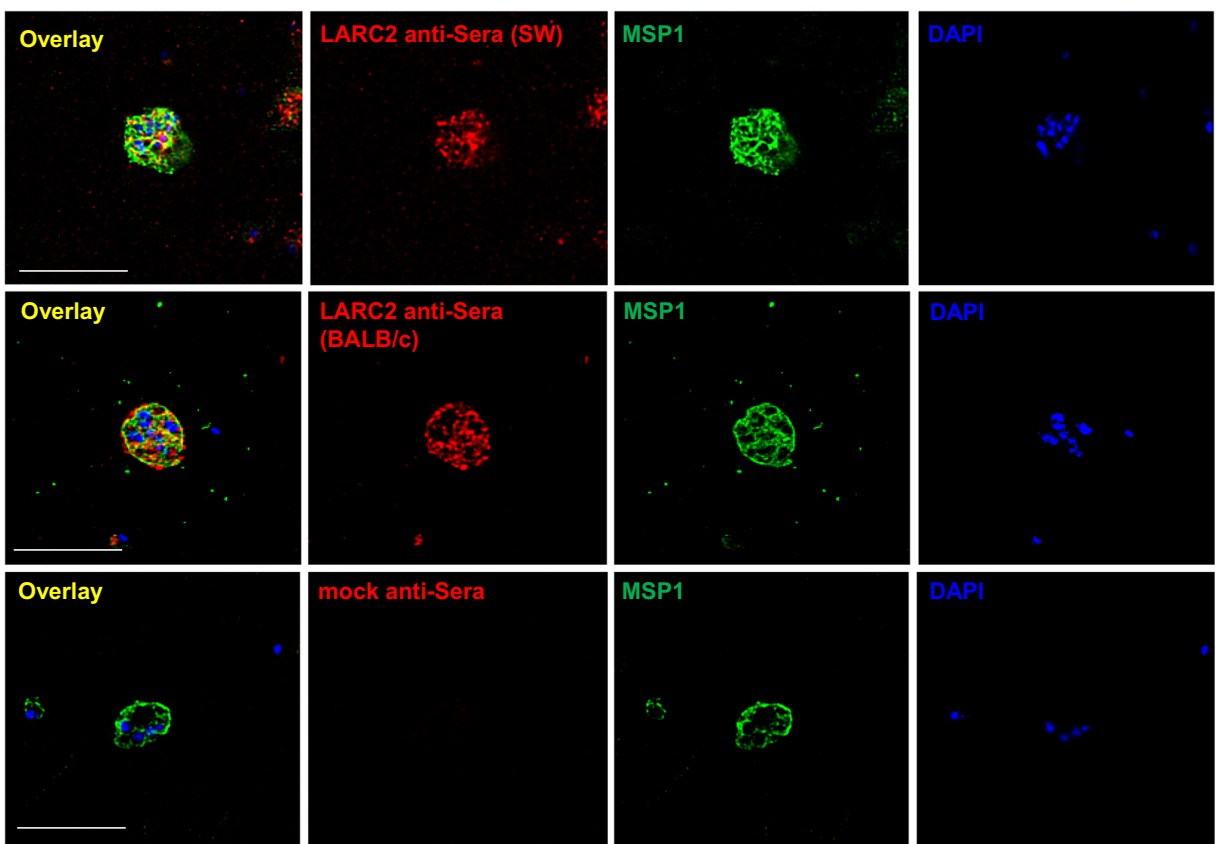

◀ **Figure EV3. PyLARC2 immunization of elicits cross-species humoral responses against *P. berghei* liver and blood stages.**

(A) Plasma was collected from six PyLARC2 immunized SW mice and pooled. The pooled plasma was then diluted 1:100 and used to evaluate reactivity of immune sera to antigens at different timepoints of *P. berghei* ANKA WT LS development at 36 hpi using IFA. Sera is shown in red, anti-mHSP70 antibody is used as control (green) and DNA is stained with DAPI (blue). Scale bar is 20 μm. In more mature *P. berghei* LS, PyLARC2 IgG displayed cross-reactivity to *P. berghei* antigens located in both the cytoplasm and the membranes of the parasite. No reactivity was observed in plasma harvested from mock-immunized mice, indicating that the staining was specific to the PyLARC2 immunized mice. (B) Plasma was collected from six PyLARC2 immunized SW or BALB/cJ mice. Pooled plasma for each mouse strain was diluted 1:100 and used to evaluate reactivity of immune sera to antigens expressed in *P. berghei* ANKA WT blood stages. Sera is shown in red, anti-MSP1antibody is used as control (green) and DNA is stained with DAPI (blue). Scale bar is 10 μm. Immune sera from both SW and BALB/cJ mice reacted with antigens expressed in the cytoplasm of blood stages. No reactivity was observed in plasma harvested from mock-immunized mice, indicating that the staining was specific to immune sera from PyLARC2 immunized mice.

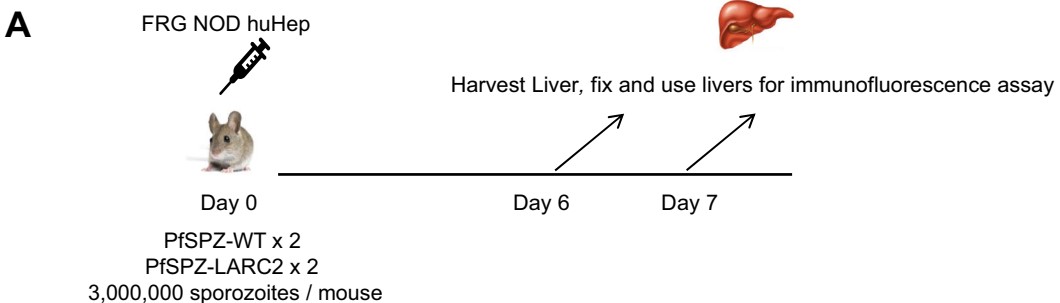

**A**

FRG NOD huHep

Harvest Liver, fix and use livers for immunofluorescence assay

Day 0

Day 6    Day 7

PfSPZ-WT x 2
PfSPZ-LARC2 x 2
3,000,000 sporozoites / mouse

**B**

| Overlay | CSP | Exp1 | DAPI |
|---|---|---|---|

PfSPZ-WT D6

PfSPZ-LARC2 D6

**C**

| Overlay | ACP | HSP70 | DAPI |
|---|---|---|---|

PfSPZ-WT D7

PfSPZ-LARC2 D7

◄ **Figure EV4. PfLARC2 LS display severe defects in late LS differentiation.**

(A) The schematic depicts the experimental design. To evaluate LS development of PfLARC2 in FRG NOD huHep mice, three million aseptic, cryopreserved sporozoites of either PfSPZ-WT or PfSPZ-LARC2 were injected intravenously into two FRG NOD huHep mice per group, respectively. Livers were harvested on days 6 and 7 post infection, fixed and liver tissue sections used for IFA analysis. LS development of PfSPZ-LARC2 was compared to PfSPZ-WT using antibodies against (B), the PPM and cytomere marker, CSP and the PVM marker, Exp1 on day six liver sections, (C) parasite mitochondrial protein, heat shock protein 70 (HSP70, green), apicoplast protein, ACP (red) on day 7 liver sections. DNA is stained with DAPI. In (B) and (C), scale bar is 20 μm. Defective cytomeres were evident on day six PfSPZ-LARC2 LS compared to PfSPZ-WT, where CSP staining was localized to invaginating PPM. PfSPZ-LARC2 LS also displayed incomplete organellar and DNA segregation.

