## [Peer Review File · EMBO Molecular Medicine]

A replication competent *Plasmodium falciparum* parasite completely attenuated by dual gene deletion

Debashree Goswami, Hardik Patel, William Betz, Janna Armstrong, Nelly Camargo, Asha Patil, Sumana Chakravarty, Sean Murphy, B. Kim Lee Sim, Ashley Vaughan, Stephen Hoffman, and Stefan Kappe

Corresponding author(s): Stefan Kappe (stefan.kappe@seattlechildrens.org)

Review Timeline:

Submission Date:	25th Jul 23
Editorial Decision:	27th Jul 23
Appeal Received:	31st Jul 23
Editorial Decision:	6th Sep 23
Revision Received:	18th Jan 24
Editorial Decision:	16th Feb 24
Revision Received:	6th Mar 24
Accepted:	7th Mar 24

Editor: Poonam Bheda

Transaction Report:

26th Jul 2023

Decision on your manuscript EMM-2023-18404

Dear Dr. Goswami,

Thank you for submitting your manuscript "A replication competent Plasmodium falciparum parasite completely attenuated by dual gene deletion" to EMBO Molecular Medicine. I have now had a chance to discuss your article with the other members of our editorial team. I am afraid that we concluded that your manuscript is not well suited for publication in EMBO Molecular Medicine and have therefore decided not to proceed with peer review. However, I have taken the liberty to also discuss your study with my colleague Eric Sawey - the executive editor of our sister journal Life Science Alliance, which publishes work that is of high value to the respective communities across all areas in the life sciences - and he would be happy to offer peer review if you were to transfer your manuscript there.

From our side, we find that the article doesn't fit well within EMBO Molecular Medicine as we focus primarily on studies that provide functional novel insights of clinical and/or translational significance in an appropriate model, but also that are conceptually novel and of broad interest. As we do not feel that this is the case here, we therefore cannot offer further consideration to your manuscript.

That being said - and as mentioned above - my colleague Eric Sawey would be pleased to send your manuscript out for formal peer-review at Life Science Alliance in its current form. You can take advantage of this offer by transferring your study to Life Science Alliance using the link below; no re-formatting is required.

I am very sorry to disappoint you on this occasion. Please rest assured that this is not a judgment of the quality or interest of your work, but a decision based on the scope requirement of our journal.

Yours sincerely,

Poonam Bheda

Poonam Bheda, PhD
Scientific Editor
EMBO Molecular Medicine

=====

As a service to authors, EMBO provides authors with the possibility to transfer a manuscript that one journal cannot offer to publish to another EMBO publication. The full manuscript and if applicable, reviewers reports are automatically sent to the receiving journal to allow for fast handling and a prompt decision on your manuscript. For more details of this service, and to transfer your manuscript to another EMBO title please click on Link Not Available

Dear Dr. Bheda,

Thank you for taking the time to consider our manuscript. **We respectfully appeal your surprising decision of rejecting our manuscript for peer-review.** In your response, you mentioned that you feel our paper does not meet three key criteria for consideration in EMBO Molecular Medicine.

Firstly, you noted that our manuscript does not provide functional novel insights of clinical and/or translational significance. This is a surprising conclusion considering that we have created the first completely attenuated, replication-competent, whole cell (via synthetically lethal dual gene deletion) human malaria parasite vaccine called LARC2. This strain is now the leading whole cell vaccine candidate for malaria and is slated for rapid clinical development as a vaccine for malaria elimination. What can be more significant than this in the infectious diseases/malaria vaccine space? We show that injectable vaccine formulations consisting of LARC2 sporozoites can be manufactured (PfSPZ-LARC2). This will allow immediate clinical development of PfSPZ-LARC2, initiating with dose-escalation safety studies in malaria naïve adults followed by a 3-dose vaccination regimen that will evaluate magnitude and durability of protection against CHMI with the Pf 7G8 strain, which is highly genetically divergent from the NF54 strain used to create Pf LARC2.

Secondly, it appears that you have concerns about the appropriateness of the models used. We want to emphasize that we employed the *Fah^{-/-}Rag2^{-/-}IL2rg^{-/-}* mice transplanted with primary human hepatocytes (FRG KO huHep) as our model system, which has been widely recognized as the only viable in vivo model for studying liver stage development of human malaria parasite species and transition to blood stage infection (Vaughan et al., Journal of Clinical Investigation 2012, Mikolajczak et al, Cell Host & Microbe, 2015, Vaughan et al, Nature Methods, 2015, etc.). In this study we clearly demonstrate in the FRGN huHep/hu RBC model that PfSPZ-LARC2 infects human hepatocytes and develops until late liver stage before undergoing complete liver stage attenuation. We also demonstrate that LARC2 fails to transition to blood stage infection, providing the most conclusive preclinical evidence possible for complete parasite attenuation at liver stage. In addition to our studies with PfSPZ LARC2, we also performed extensive immunization/challenge studies with LARC2 in the rodent malaria model, *P. yoelii* LARC2 (Py LARC2), to evaluate the safety and efficacy of the LARC2 vaccine. We clearly demonstrate that LARC2 vaccine elicits robust and broad CD8⁺ T cell-based protective immunity. Our study therefore used two key models (the FRGN model for Pf LARC2 and generic research mice for Py LARC2) to demonstrate attenuation, immunogenicity and efficacy of the LARC2 vaccine strain.

Thirdly, we believe our findings are conceptually novel and of broad interest. With LARC2, we have created the first completely attenuated, replication-competent (via synthetically lethal dual gene deletion) human malaria parasite vaccine. We show that although each individual gene deletion does not result in complete parasite attenuation, the simultaneous deletion of both genes does result in complete attenuation at late liver stage. Based on all available evidence, the PfSPZ-LARC2 vaccine thus represents the most promising malaria vaccine candidate to prevent malaria infection and as such, our research findings are highly relevant and of great interest to the global public health-, vaccine- and infectious diseases- research communities.

Lastly, I would like to point out that I have served as a reviewer for EMBO Molecular Medicine in the space of live-attenuated malaria vaccine research and that your journal had send out papers for peer review in the past that are of much less significance and impact compared to the work we present in our manuscript. With these points in mind, we kindly request a reconsideration of our manuscript for peer-review.

With best regards

Stefan Kappe

Stefan Kappe, PhD

Professor and Associate Vice Chair for Basic Research

Department of Pediatrics, University of Washington

Associate Director

Center for Global Infectious Disease Research

Seattle Children's Research Institute

O: 206.884.3180 | C: 206.310.9508

6th Sep 2023

Dear Dr. Goswami,

Thank you for the submission of your manuscript to EMBO Molecular Medicine. We have now received feedback from the three reviewers who agreed to evaluate your manuscript. As you will see from the reports below, the referees acknowledge the interest of the study and are overall supporting publication of your work pending appropriate revisions.

Addressing the reviewers' concerns in full will be necessary for further considering the manuscript in our journal, and acceptance of the manuscript will entail a second round of review. Please note that editorially and upon discussion with the referees, we agree that experimental comparison between late-arresting and early-arresting GAPs as suggested by Reviewer 3 is not necessary for further consideration at EMM; however this point should be addressed by discussion. All other reviewer concerns should be fully addressed.

EMBO Molecular Medicine encourages a single round of revision only and therefore, acceptance or rejection of the manuscript will depend on the completeness of your responses included in the next, final version of the manuscript. For this reason, and to save you from any frustrations in the end, I would strongly advise against returning an incomplete revision.

We are expecting your revised manuscript within three months, if you anticipate any delay, please contact us.

We require:

4) A .docx formatted letter INCLUDING the reviewers' reports and your detailed point-by-point responses to their comments. As part of the EMBO Press transparent editorial process, the point-by-point response is part of the Review Process File (RPF), which will be published alongside your paper.

5) A complete author checklist, which you can download from our author guidelines (<https://www.embopress.org/page/journal/17574684/authorguide#submissionofrevisions>). Please insert information in the checklist that is also reflected in the manuscript. The completed author checklist will also be part of the RPF.

6) Please note that all corresponding authors are required to supply an ORCID ID for their name upon submission of a revised manuscript.

7) It is mandatory to include a 'Data Availability' section after the Materials and Methods. Before submitting your revision, primary datasets produced in this study need to be deposited in an appropriate public database, and the accession numbers and database listed under 'Data Availability'. Please remember to provide a reviewer password if the datasets are not yet public (see <https://www.embopress.org/page/journal/17574684/authorguide#dataavailability>).

In case you have no data that requires deposition in a public database, please state so in this section. Note that the Data Availability Section is restricted to new primary data that are part of this study. This study includes no data deposited in external repositories.

8) For data quantification: please specify the name of the statistical test used to generate error bars and P values, the number (n) of independent experiments (specify technical or biological replicates) underlying each data point and the test used to calculate p-values in each figure legend. The figure legends should contain a basic description of n, P and the test applied. Graphs must include a description of the bars and the error bars (s.d., s.e.m.). Please provide exact p values.

13) Author contributions: CRediT has replaced the traditional author contributions section because it offers a systematic machine readable author contributions format that allows for more effective research assessment. Please remove the Authors Contributions from the manuscript and use the free text boxes beneath each contributing author's name in our system to add specific details on the author's contribution. More information is available in our guide to authors.

Share synopsis text and image, as well as eTOC:

Please note that these would be the final versions and changes during proofing are usually not allowed

16) As part of the EMBO Publications transparent editorial process initiative (see our Editorial at <http://embomolmed.embopress.org/content/2/9/329>), EMBO Molecular Medicine will publish online a Review Process File (RPF) to accompany accepted manuscripts.

In the event of acceptance, this file will be published in conjunction with your paper and will include the anonymous referee reports, your point-by-point response and all pertinent correspondence relating to the manuscript. Let us know whether you agree with the publication of the RPF and as here, if you want to remove or not any figures from it prior to publication. Please note that the Authors checklist will be published at the end of the RPF.

EMBO Molecular Medicine has a "scooping protection" policy, whereby similar findings that are published by others during

review or revision are not a criterion for rejection. Should you decide to submit a revised version, I do ask that you get in touch after three months if you have not completed it, to update us on the status.

I look forward to receiving your revised manuscript.

Yours sincerely,

Poonam Bheda

Poonam Bheda, PhD
Scientific Editor
EMBO Molecular Medicine

***** Reviewer's comments *****

Referee #1 (Comments on Novelty/Model System for Author):

This manuscript describes the generation of late-arresting Plasmodium parasites that can be used for vaccination against malaria. The authors begin by validating their proposed gene deletion using the *P. yoelii* rodent model and then go on to generate equivalent deletions in *P. falciparum*. They convincingly demonstrate that PfSPZ-LARC2's hepatic growth arrests before the release of infectious merozoites, validating it as a vaccine candidate against this human parasite. Although the strategy in itself is not entirely novel, the rodent model system employed is perfectly adequate and the potential medical impact of an efficacious late-arresting vaccine against malaria is enormous.

Referee #1 (Remarks for Author):

This is a very interesting report, whose suitability for publication is beyond dispute. The authors conducted a well-structured study, their conclusions are supported by the data, and the manuscript is well-written, its message being clear even to the non-specialized reader. I therefore have no objections to the publication of this manuscript. My only suggestion/remark, is that it would have been interesting to conduct the growth arrest and immunization experiments in mice using cryopreserved *P. yoelii* LARC2 parasites, as opposed to sporozoites collected fresh from mosquito salivary glands, since that will be the proposed formulation for human immunizations with PfSPZ-LARC2. Other than that, I have no further remarks on this nice manuscript.

Referee #2 (Remarks for Author):

To the authors,

This is a remarkable work that could lead the way of designing vaccines based on GMOs. Current vaccines based on irradiated parasites don't offer a perfect mimic of what a wild type parasite would be, as these may contain modifications on many different antigens to obtain fully attenuated parasites. Bypassing this drawback by designing a Plasmodium line with arrested development, but otherwise equal to the real infections, has a lot of potential to be an effective vaccine, as discussed and well proven in this manuscript. I would like to support the publication of this work, and I hope to see soon the next steps it takes. Said this, I have some questions and suggestions, one of them might require experimental work to answer, but hopefully won't require big experimental design or time.

1. When exploring how the double knock-out PfLARC2 parasites developed in the mosquito, 3 biological replicates are represented. It is stated the number of mosquitos dissected to count oocyst per midgut (12 per cage) but it is not mentioned if the number of mosquitos were the same to check the sporozoite numbers. The number of mosquitoes dissected, to check the development of the PyLARC2 is not stated either.
2. On the matter of in which mechanisms does the immunity of the mice relies on; the IgGs from plasma of the immunized mice were found reactive against parasite samples in IFA, both pre and erythrocytic stages.
 - a. To my understanding, these IFAs were prepared with samples from *P. yoelii* WT, are they? It is not stated in the main text, nor in Figure S3 caption, please add this information for clarity.
 - b. Do these IgGs in plasma have reactivity against *P. berghei* too? This might provide hints on whether if antibodies could also have potential contribution in this cross-species immunity. IFA against *P. berghei* samples can be informative. Alternatively, a western blot with parasite extracts from blood from both Py and Pb, and readouts using plasma of immunized mice as primary antibody, could be used for the same purpose.
3. To explore safety issues before starting experiments with humans, have you consider immunization in non-human primates first?

Referee #3 (Comments on Novelty/Model System for Author):

The manuscript presents the construction and validation of double knockout *P. yoelii* and *P. falciparum* sporozoites that arrest later in intrahepatic development than previously reported mutants. The *P. yoelii* mutant was validated in rodent models and the *P. falciparum* mutant is validated in the humanized mouse model. Both models are well established in the literature. Late-arresting liver stages are expected to express both liver stage and blood stage antigens, unlike early arresting liver stages that have a smaller antigenic repertoire and lower parasite mass. Immunization with these mutant sporozoites is hypothesized to elicit "better" protection than early-arresting liver stage parasites and offer protection against both sporozoite and blood stage infection.

The construction of the *P. falciparum* mutants is technically not facile and their validation in humanized mouse models offers novelty. However, the work offers limited conceptual advances and results do not demonstrate a significant improvement in protection compared to previously reported results using early-arresting liver stages. A key experiment, that is absent, would be to test early-arresting and late-arresting parasites in parallel to determine differences, if any, in protection conferred by them. As it stands the manuscript's insights are of interest to a niche audience and therefore it is more suitable for a specialized journal.

Referee #3 (Remarks for Author):

Goswami et al describe the construction of *P. yoelii* and *P. falciparum* mutants for use as "late arresting liver stages" for use in whole-sporozoite immunization against malaria. The parasites missing 2 proteins that are required for liver stage development. The expectation is that these late-arresting liver stages will elicit "longer lasting" protection as well as protect against both sporozoites and blood stages. These hypotheses are tested in both *P. yoelii* and *P. falciparum* mutants. Generating *P. falciparum* sporozoites missing 2 genes is a significant challenge, with only a handful of existing *P. falciparum* mutant lines making it to sporozoites. Therefore the manuscript represents a technical achievement.

However, it is less novel and limited in its insights - the hypothesis being tested ie late-arresting liver stages will protect against pre-erythrocytic and erythrocytic stage challenges, is well established. The manuscript does not move the needle in identifying possible antigens present in late liver stages but absent in early liver stages, or more specific mechanisms responsible for this pan-stage protection.

In addition, some of the results are puzzling.

- 1) There should be a clear definition of what exactly constitutes an "early arresting" liver stage versus a "late arresting" liver stage. Since size of WT and PfLARC2 is similar, size is not a surrogate marker for development. What is the expression of MTIP and MSP1 in early arresting liver stages?
- 2) Has there been parallel testing of early-arresting and late-arresting GAPs to compare the duration and cross-strain protection elicited by them? This is a central question since it goes to the rationale underlying the generation of the LARCs ie LARCs will elicit more durable immunity than early-arresting GAPs that have been previously tested.
- 3) Despite the lack of major merozoite proteins (that are also expressed in erythrocytic stages) such as MSP1 and MTIP in PyLARC2 (also either absent or weakly expressed in PfLARC2), immunization with PyLARC2 protects against a blood stage challenge and sera from immunized animals reacts against WT blood stage merozoites in a pattern that appears to overlap with MSP1 and MTIP (Supp Fig 3). What is the mechanism of the protection against blood stages, if major blood stage antigens are absent in LARCs?
- 4) The delay in patency in S/W mice (Fig 3A) should be reported.
- 5) In Fig 2B, i.m. injections show some protection. As the authors note their results contrast with PMID but do not discuss potential differences that could be responsible for the disparity. It would be useful to include such a discussion since efficacy through i.m. administration is important for a vaccine.

Minor comments:

Methods should include source of all antibodies used in IFAs.

Why does Fig 1 utilize a multiple comparison test - there are only 2 groups (WT and PfLARC2)?

Response to Reviewers

We like to thank reviewers for their positive comments and constructive critiques, which have helped us to strengthen the manuscript. We have revised the manuscript in response to the comments and added additional data where possible. Major changes to the revised manuscript are marked in yellow. Below, please find the point-by-point responses to each reviewers' critique.

Referee 1 (Comments on Novelty/Model System for Author):

This manuscript describes the generation of late-arresting Plasmodium parasites that can be used for vaccination against malaria. The authors begin by validating their proposed gene deletion using the *P. yoelii* rodent model and then go on to generate equivalent deletions in *P. falciparum*. They convincingly demonstrate that PfSPZ-LARC2's hepatic growth arrests before the release of infectious merozoites, validating it as a vaccine candidate against this human parasite. Although the strategy in itself is not entirely novel, the rodent model system employed is perfectly adequate and the potential medical impact of an efficacious late-arresting vaccine against malaria is enormous.

→ We thank the reviewer for the positive feedback on the manuscript.

Referee #1 (Remarks for Author):

1. This is a very interesting report, whose suitability for publication is beyond dispute. The authors conducted a well-structures study, their conclusions are supported by the data, and the manuscript is well-written, its message being clear even to the non-specialized reader. I therefore have no objections to the publication of this manuscript. My only suggestion/remark, is that it would have been interesting to conduct the growth arrest and immunization experiments in mice using cryopreserved *P. yoelii* LARC2 parasites, as opposed to sporozoites collected fresh from mosquito salivary glands, since that will be the proposed formulation for human immunizations with PfSPZ-LARC2. Other than that, I have no further remarks on this nice manuscript.

→ We thank the reviewer for the positive feedback on the manuscript. In response to the suggestion by the reviewer, we have now conducted immunizations of mice with cryopreserved *P. yoelii* SPZ-LARC2 parasites. We immunized BALB/c mice with two doses of 50,000 cryopreserved PyLARC2 sporozoites, two weeks apart and challenged them with 10,000 fully infectious cryopreserved PyXNL wildtype sporozoites, two weeks after the final boost. All *P. yoelii* LARC2 immunized mice were protected, as shown below. The

data have been incorporated into the revised manuscript (new Figure 3B, new Table 4 and Lines 248 – 254).

Immunization with cryopreserved Py LARC2 protects against pre-erythrocytic stage challenge.

Mouse Strain	Parasite Genotype	Route of Immunization	Number of sporozoites used for immunization/challenge			Number of mice protected (days to patency)
			Prime	Boost (days after prime)	Challenge (days after boost)	
BALB/c	NA	NA	-	-	10,000 (14)	0/6 (5)
	Cryopreserved PySPZ-LARC2	i.v.	50,000	50,000 (15)	10,000 (14)	12/12

Referee 2 (Remarks for Author):

To the authors,

This is a remarkable work that could lead the way of designing vaccines based on GMOs. Current vaccines based on irradiated parasites don't offer a perfect mimic of what a wild type parasite would be, as these may contain modifications on many different antigens to obtain fully attenuated parasites. Bypassing this drawback by designing a Plasmodium line with arrested development, but otherwise equal to the real infections, has a lot of potential to be an effective vaccine, as discussed and well proven in this manuscript. I would like to support the publication of this work, and I hope to see soon the next steps it takes. Said this, I have some questions and suggestions, one of them might require experimental work to answer, but hopefully won't require big experimental design or time.

→ We thank the reviewer for the positive feedback on the manuscript.

1. When exploring how the double knock-out PfLARC2 parasites developed in the mosquito, 3 biological replicates are represented. It is stated the number of mosquitos dissected to count oocyst per midgut (12 per cage) but it is not mentioned if the number of mosquitos were the same to check the sporozoite numbers. The number of mosquitoes dissected, to check the development of the PyLARC2 is not stated either.

→ We have determined the count of oocysts per midgut as well as the salivary gland sporozoites from the same infected mosquito cages. At least 50 mosquitoes were dissected

for the quantification of salivary gland sporozoites/mosquito for both Pf LARC2 as well as Py LARC2. The materials and methods have been updated to provide clarity for this.

2. On the matter of in which mechanisms does the immunity of the mice relies on; the IgGs from plasma of the immunized mice were found reactive against parasite samples in IFA, both pre and erythrocytic stages.

- a) To my understanding, these IFAs were prepared with samples from *P. yoelii* WT, are they? It is not stated in the main text, nor in Figure S3 caption, please add this information for clarity.

→ Yes, the liver stage and blood stage IFAs were performed with Py WT infected livers and infected red blood cells, respectively. The main text, figure legend and materials and methods have been updated to provide clarity for this.

- b) Do these IgGs in plasma have reactivity against *P. berghei* too? This might provide hints on whether if antibodies could also have potential contribution in this cross-species immunity. IFA against *P. berghei* samples can be informative. Alternatively, a western blot with parasite extracts from blood from both Py and Pb, and readouts using plasma of immunized mice as primary antibody, could be used for the same purpose.

→ We thank the reviewer for this suggestion. To analyze potential cross-species reactivity of Py LARC2 immune sera to the *P. berghei* liver stages, we infected SW mice with 250,000 *P. berghei* ANKA sporozoites, harvested livers at 36 hpi, followed by fixation and processing for IFA. Primary antibodies used were Py LARC2-immune sera and anti-parasite mHSP70 antibody as control. Similarly, to analyze cross-species reactivity of Py LARC2 immune sera to *P. berghei* blood stages, *P. berghei* infected red blood cells were fixed and processed for IFA. Primary antibodies used were Py LARC2-immune sera from either SW or BALB/c mice and MSP1 antibody as control. As shown below and in the revised manuscript, Py LARC2 immune sera from SW mice recognized antigens of *P. berghei* liver stages (Figure EV 3A), which is similar to what we observed for Py liver stages (Figure EV2A). Furthermore, immune sera from PyLARC2 immunized BALB/c and SW mice reacted with antigens expressed in *P. berghei* blood stage parasites (Figure EV3A). Sera from mock-immunized mice did not show any reactivity against *P. berghei* liver stage or blood stage antigens, indicating that the observed reactivity was specific (Figure EV3A-B). These results show that the PyLARC2 mediated pre-erythrocytic stage immunity against a *P. berghei* sporozoite challenge might be attributable to antibodies elicited against antigens shared between *P. berghei* and Py liver stages and blood stages. The data have been incorporated in the revised manuscript and results are described with lines 310 - 322.

A

B

3. To explore safety issues before starting experiments with humans, have you consider immunization in non-human primates first?

→ We thank the reviewer for the feedback. Indeed, we considered immunizations in non-human primates. However, we opted against this approach for several reasons. Firstly, rhesus macaques are less susceptible to pre-erythrocytic stages of Pf NF54, making them suboptimal models for assessing blood stage breakthroughs associated with PfSPZ-LARC2 (Voinson, Nunn, and Goldberg 2022). Secondly, although Aotus sp. monkeys could have been an alternative, regrettably, we lack access to these primates. Importantly, in our past work we tested the PfGAP3KO strain in FRG mice and after observing no breakthrough blood stage infection, we went straight to human CHMI studies (Kublin et al. 2017; Mikolajczak et al. 2014). Therefore, we think it is unnecessary to conduct safety studies in non-human primates before human safety trials.

Referee #3 (Comments on Novelty/Model System for Author):

The manuscript presents the construction and validation of double knockout *P. yoelii* and *P. falciparum* sporozoites that arrest later in intrahepatic development than previously reported mutants. The *P. yoelii* mutant was validated in rodent models and the *P. falciparum* mutant is validated in the humanized mouse model. Both models are well established in the literature. Late-arresting liver stages are expected to express both liver stage and blood stage antigens, unlike early arresting liver stages that have a smaller antigenic repertoire and lower parasite mass. Immunization with these mutant sporozoites is hypothesized to elicit "better" protection than early-arresting liver stage parasites and offer protection against both sporozoite and blood stage infection.

The construction of the *P. falciparum* mutants is technically not facile and their validation in humanized mouse models offers novelty. However, the work offers limited conceptual advances and results do not demonstrate a significant improvement in protection compared to previously reported results using early-arresting liver stages. A key experiment, that is absent, would be to test early-arresting and late-arresting parasites in parallel to determine differences, if any, in protection conferred by them. As it stands the manuscript's insights are of interest to a niche audience and therefore it is more suitable for a specialized journal.

→ Based on feedback from the editor, we did not perform the comparative experiments requested by the reviewer. We would also like to point out that this question has been addressed by us in the past (Butler et al. 2011). This study found rodent malaria LARC GAPs to be more efficacious than EARD GAPs and radiation-attenuated sporozoites.

Referee #3 (Remarks for Author):

Goswami et al describe the construction of *P. yoelii* and *P. falciparum* mutants for use as "late arresting liver stages" for use in whole-sporozoite immunization against malaria. The parasites missing 2 proteins that are required for liver stage development. The expectation is that these late-arresting liver stages will elicit "longer lasting" protection as well as protect against both sporozoites and blood stages. These hypotheses are tested in both *P. yoelii* and *P. falciparum* mutants. Generating *P. falciparum* sporozoites missing 2 genes is a significant challenge, with only a handful of existing *P. falciparum* mutant lines making it to sporozoites. Therefore, the manuscript represents a technical achievement.

1. However, it is less novel and limited in its insights - the hypothesis being tested i.e. late-arresting liver stages will protect against pre-erythrocytic and erythrocytic stage challenges, is well established. The manuscript does not move the needle in identifying possible antigens present in late liver stages but absent in early liver stages, or more specific mechanisms responsible for this pan-stage protection.

The significance of our manuscript lies in the creation of Pf LARC2, the first completely attenuated, replication-competent, whole cell human malaria vaccine strain currently undergoing clinical development. This strain is now the leading whole cell vaccine candidate for malaria. What can be more significant than this in the infectious diseases/malaria vaccine space? We show that injectable vaccine formulations consisting of LARC2 sporozoites can be manufactured (PfSPZ-LARC2). Additionally, we have also performed extensive pre-clinical evaluation to demonstrate the infectivity and safety profile of PfSPZ-LARC2 in humanized mice. Thus, PfSPZ-LARC2 represents the next generation of whole sporozoite vaccines against malaria, intended for licensure. We agree that the identification of liver stage antigens that are expressed in LARC GAPs is of great importance, but our work focuses on the development of a safe and potent whole cell malaria vaccine. As such, antigen discovery is beyond the scope of this manuscript.

2. There should a clear definition of what exactly constitutes an "early arresting" liver stage versus a "late arresting" liver stage. Since size of WT and PfLARC2 is similar, size is not a surrogate marker for development. What is the expression of MTIP and MSP1 in early arresting liver stages?

→ Regarding the reviewers first point on what exactly constitutes an "early arresting" liver stage versus a "late arresting" liver stage, this has been comprehensively reviewed (Goswami, Minkah, and Kappe 2019). In brief, an early liver stage-arresting GAP can successfully invade hepatocytes and transform into LS trophozoites but does not at all or only undergoes very early LS schizogony (2-4 N genomes). Thus, we have termed these

GAPs early liver stage-arresting replication deficient (EARD). Radiation attenuated sporozoites that arrest early in LS development can also be classified as EARD. In contrast, late liver stage-arresting replication competent (LARC) parasites undergo massive DNA and organellar replication (10,000+ N genome replication) and significant expansion of parasite cell biomass. Based on our comprehensive exploration of Pf LS development in humanized mice for Pf WT and Pf GAPs, we have noted substantial impairments in hallmarks of late liver stage differentiation in LARC GAPs. These deficiencies encompass impaired DNA and organellar segregation, absence of cytomere formation and importantly, lack of formation of mature liver stage merozoites (Goswami et al. 2020; Goswami, Arredondo, et al. 2022; Goswami, Kumar, et al. 2022). MSP1 is not expressed in EARD GAPs and this is expected as it is not expressed in early liver stages. MTIP on the other hand is still present in early liver stage trophozoites but not in LARC2 GAP late liver stages.

We have modified the revised manuscript, such that the lines 82 – 85 in the manuscript and lines 106 – 108, clearly describe what constitutes EARD whole parasites and LARC GAPs.

82 – 85: PfSPZ Vaccine is the prototype of early liver stage-arresting replication-deficient (EARD) whole parasite vaccines; PfSPZ invade hepatocytes but arrest early in liver stage development before significant initiation of genome replication and differentiation (LS schizogony) due to radiation-induced DNA damage.

106 – 108: LARC GAPs undergo extensive LS DNA and organellar replication and cell mass expansion, but developmentally arrest before differentiation into mature LS merozoites (34,45).

3. Has there been parallel testing of early-arresting and late-arresting GAPs to compare the duration and cross-strain protection elicited by them? This is a central question since it goes to the rationale underlying the generation of the LARCs ie LARCs will elicit more durable immunity than early-arresting GAPs that have been previously tested.
→ Based on feedback from the editor, we did not perform the comparative experiments requested by the reviewer. We would also like to point out that this question has been addressed by us in the past (Butler et al. 2011). This study found rodent malaria LARC GAPs to be more efficacious than EARD GAPs and radiation-attenuated sporozoites.
4. Despite the lack of major merozoite proteins (that are also expressed in erythrocytic stages) such as MSP1 and MTIP in PyLARC2 (also either absent or weakly expressed in PflARC2), immunization with PyLARC2 protects against a blood stage challenge and sera from immunized animals reacts against WT blood stage merozoites in a pattern that appears to overlap with MSP1 and MTIP (Supp Fig 3). What is the mechanism of the protection against blood stages, if major blood stage antigens are absent in LARCs?

→ We have previously demonstrated that immunization with LARC GAP (Py *fabb/f⁻* parasites) elicits robust T- and B-cell responses that provide stage-transcending protection. However, the importance of each immune component depends on the genetic background of the mouse strains used (Sack et al. 2015). Notably, in C57BL/6 mice, antibody-mediated responses protected against Py blood stage challenges. Conversely, Balb/c mice exhibited a greater dependence on CD4/CD8 T cell-mediated protection, evidenced by the loss of protection to lethal Py blood stage challenges upon depletion of CD4/CD8+ T cells. These findings suggest that the protective mechanism of PyLARC2 against Py blood stage infection in Balb/c mice is likely contingent on the CD4/CD8+ T cell responses generated against antigens expressed during both liver and blood stage infections. We have elaborated on this aspect in the manuscript (Lines 574 – 580).

Lines 576 – 582: In support of this notion, we show that immune sera from LARC2-immunized mice reacted with late LS and blood stages. However, whether this humoral stage-transcending reactivity contributes to protection against blood stages remains unknown. Interestingly, we have previously observed that Balb/c mice immunized with the Py *fabb/f⁻* LARC GAP loose protection to lethal Py blood stage challenge upon depletion of CD4/CD8+ T cells, but they did not substantially contribute to protect C57BL/6 mice, in which protection was largely antibody dependent (85).

5. The delay in patency in S/W mice (Fig 3A) should be reported.

→ The following sentences have been modified (Lines 242 – 247).

Lines 242 – 247: All naive and mock-immunized mice developed blood stage infection four days after sporozoite infection (Figure 3A, Table 3). Seven of ten PyLARC2-immunized mice exhibited sterile protection against sporozoite challenge, while the three mice that were blood stage positive showed a longer pre-patent period compared to naïve and mock infected mice, with a two-day delay in patency, indicating partial protection in these mice.

6. In Fig 2B, i.m. injections show some protection. As the authors note their results contrast with PMID but do not discuss potential differences that could be responsible for the disparity. It would be useful to include such a discussion since efficacy through i.m. administration is important for a vaccine.

→ We completely agree with the reviewer that this is an important advancement achieved by the PyLARC2 GAP vaccine, which was difficult to achieve with EARD sporozoite vaccines,

including with the gold standard radiation attenuated sporozoites vaccine. We have now expanded our discussion of this point in the revised manuscript (Lines 538 – 545)

Lines 540 – 547: Importantly, we observed that i.m. immunization with two doses of 20,000 PyLARC2 sporozoites elicited 70% sterilizing protection against high dose i.v. sporozoite challenge, a result that cannot be achieved with similar doses of EARD vaccines (67,77–79). It has been shown that i.m. administered sporozoites home to the liver less effectively, reducing the hepatic immunogen load that can prime protective CD8 T cell responses (79–81). PyLARC2 LS replicate and undergo massive cell expansion, which in turn increases antigen biomass within the infected hepatocytes. This might in part explain the good efficacy observed herein with LARC2 i.m. vaccination.

Minor comments:

1. Methods should include source of all antibodies used in IFAs.
→ Source of all antibodies used for IFAs is now mentioned in the materials and methods.

2. Why does Fig 1 utilize a multiple comparison test - there are only 2 groups (WT and PflARC2)?
→ In Fig 1A, the mean of multiple groups is compared, including;
 - a. Mean of WT at 36 hpi compared to mean of LARC2 36 hpi
 - b. Mean of WT at 36 hpi compared to the mean of WT at 48 hpi
 - c. Mean of WT at 36 hpi compared to the mean of LARC2 at 48 hpi
 - d. Mean of LARC2 at 36 hpi compared to the mean of LARC2 at 48 hpi

These comparisons were not shown in the figure to avoid confusion and only the relevant comparisons were included. The figure has now been updated to include all comparisons.

16th Feb 2024

Dear Prof. Kappe,

Thank you for the submission of your revised manuscript to EMBO Molecular Medicine. We have now received the enclosed reports from the referees that were asked to re-assess it. As you will see the reviewers are now fully supportive and I am pleased to inform you that we will be able to accept your manuscript pending the following final amendments:

1) In the main manuscript file, please do the following:

- Reduce keywords to max. 5.

- Author contributions: Please remove from the manuscript and only specify author contributions in our submission system. CRediT has replaced the traditional author contributions section because it offers a systematic machine-readable author contributions format that allows for more effective research assessment. You are encouraged to use the free text boxes beneath each contributing author's name to add specific details on the author's contribution. More information is available in our guide to authors: <https://www.embopress.org/page/journal/17574684/authorguide#authorshipguidelines>

- Please correct the reference citations in the reference list - these should not be numerical but rather alphabetical. Where there are more than 10 authors on a paper, note that only 10 will be listed, followed by "et al.". DOIs should be removed for all publications. Please check "Author Guidelines" for more information.

<https://www.embopress.org/page/journal/17574684/authorguide#referencesformat>

- Data not shown: We do not allow statements/conclusions with "data not shown". As per our guidelines, on "Unpublished Data" the journal does not permit citation of "Data not shown". All data referred to in the paper should be displayed in the main or Expanded View figures. Please remove from page 24.

2) In the Materials and Methods, please take care of the following:

- Cell lines: You have indicated in the Author Checklist that you used cell lines in the manuscript. Please include all information requested in the author checklist (accession number in repository or supplier name, catalog number, clone number, and/or RRID). Please also be sure to include a sentence in the Materials and Methods as to whether or not the cell lines were recently authenticated and tested for mycoplasma contamination - if this is not possible, please explain why in your point-by-point response to these requests.

- The Materials & Methods section on *P. falciparum* parasite lines appears to be missing some details/incomplete (at the beginning there is the following "*P. falciparum* parasite lines and were cultured...")

- Although you have indicated in the Author Checklist that "No blinding was done" has been included in the manuscript, this does not appear to be in the Experimental Animals section. Please ensure that a statement on whether or not blinding was done is included in the Materials & Methods even if no blinding was done.

- Antibodies: please ensure that company name, catalog number, and dilutions/amounts of each antibody are reported in the Materials & Methods. Currently only catalog and clone numbers are given for Flow Cytometry and some of the antibody information is missing from the Immunofluorescence assay section

3) Please place individual sections of the manuscript in the following order: Title page - Abstract & Keywords - Introduction - Results - Discussion - Materials & Methods - Data Availability - Acknowledgements - Disclosure and Competing Interests Statement - The Paper Explained - For More Information - References - Figure Legends - Expanded View Figure Legends.

4) Please move the Materials & Methods after the Discussion.

5) Please incorporate the 'Study Approval' section into the Materials & Methods.

6) The Paper Explained: Please add "The Paper Explained" to the main manuscript text.

7) For more information: This space should be used to list relevant web links for further consultation by our readers. Could you identify some relevant ones and provide such information as well? Some examples are patient associations, relevant databases, OMIM/proteins/genes links, author's websites, etc...

8) For the figures and figure legends, please take care of the following:

- Please upload the main figures and Expanded View figures as individual, high resolution figure files. Check the "Author Guidelines" for more information:

<https://www.embopress.org/page/journal/17574684/authorguide#figureformat>

- Please note that Panel F is not labelled in Figure 6

- Appendix Figure S2 should be after Appendix Figure S1 in the Appendix file

- Please note that a separate 'Data Information' section is required in the legends of figures 3c; 5a-e; 6b-c; EV 1b-e; EV 4b-c.

- Please indicate the statistical test used for data analysis in the legend of figure 7c.

- Please note that in figures 1a; 6b-c; there is a mismatch between the annotated p values in the figure legend and the annotated p values in the figure file that should be corrected.

- Please note that information related to n is missing in the legends of figure 7c; EV 1b-e.

- Please note that the error bars are not defined in the legends of figure 7c; EV 1b-e.

- Please note that we require exact p-values to be reported. Currently exact p-values are not provided.

9) Please ensure that all funding sources are entered into the manuscript submission system - it is unclear whether the National Institute of Allergy and Infectious Diseases (NIAID) be added to the list of funders.

10) Synopsis image: the resolution of the synopsis image is not high. Is this sufficient for you? Otherwise please reupload in the same dimensions but with higher resolution.

11) Synopsis text: Please check your synopsis text before submission with your revised manuscript. Please be aware that in the proof stage minor corrections only are allowed (e.g., typos).

- Please use the passive voice (edit last sentence of standfirst).

- Please check the first bullet point - should both "exhibited" and "suffered" be included?).

- Please check the formulation of the second bullet point (grammar seems awkward)

12) Appendix file: Please upload the Appendix as a single PDF and add page numbers to the Table of Contents.

13) As part of the EMBO Publications transparent editorial process initiative (see our policy here:

https://www.embopress.org/transparent-process#Review_Process), EMBO Molecular Medicine will publish online a Peer Review File (PRF) to accompany accepted manuscripts. This file will be published in conjunction with your paper and will include the anonymous referee reports, your point-by-point response and all pertinent correspondence relating to the manuscript. Let us know whether you agree with the publication of the PRF and as here, if you want to remove or not any figures from it prior to publication. Please note that the Authors checklist will be published at the end of the PRF.

14) Please provide a point-by-point letter INCLUDING my comments as well as the reviewer's reports and your detailed responses (as Word file).

I look forward to reading a new revised version of your manuscript as soon as possible.

Yours sincerely,

Poonam Bheda

Poonam Bheda, PhD
Scientific Editor
EMBO Molecular Medicine

***** Reviewer's comments *****

Referee #1 (Comments on Novelty/Model System for Author):

The model employed is highly appropriate and the potential medical impact of an effective LARC vaccine against malaria is enormous. Novelty is graded "Medium" because the notion of employing genetically-attenuated Plasmodium sporozoites as immunization agents is not, in itself, new, but the present report constitutes a remarkable advancement in this field.

Referee #1 (Remarks for Author):

The authors have done an absolutely remarkable job at addressing my (and other) comments. They did not shy away from performing additional experiments, and the added-value of the new data they obtained in this process is beyond dispute. I have no hesitations in recommending this article for publication in its current format.

Referee #2 (Remarks for Author):

Thanks to the authors for their answers. I consider that they have satisfactorily answered the comments and provided appropriate evidence for the reviewing process. This manuscript version is suitable for publication, in my opinion.

We sincerely appreciate the referees for their constructive feedback on our manuscript. Your insights have been invaluable in improving the quality and clarity of our work.

***** Reviewer's comments *****

Referee #1 (Comments on Novelty/Model System for Author):

The model employed is highly appropriate and the potential medical impact of an effective LARC vaccine against malaria is enormous. Novelty is graded "Medium" because the notion of employing genetically-attenuated *Plasmodium* sporozoites as immunization agents is not, in itself, new, but the present report constitutes a remarkable advancement in this field.

➤ We thank the reviewer for the enthusiastic comment.

Referee #1 (Remarks for Author):

The authors have done an absolutely remarkable job at addressing my (and other) comments. They did not shy away from performing additional experiments, and the added-value of the new data they obtained in this process is beyond dispute. I have no hesitations in recommending this article for publication in its current format.

➤ We thank the reviewer for the positive comment and for recommending the manuscript for publication.

Referee #2 (Remarks for Author):

Thanks to the authors for their answers. I consider that they have satisfactorily answered the comments and provided appropriate evidence for the reviewing process. This manuscript version is suitable for publication, in my opinion.

➤ We thank the reviewer for recommending for publication.

We thank the editors for their corrections and have addressed all the issues raised as mentioned below. Revised text has been underlined in the manuscript.

1) In the main manuscript file, please do the following:

- a) Reduce keywords to max. 5.
 - Key words were reduced to 5, *Malaria*, *Plasmodium*, genetically attenuated parasite vaccine, sporozoite, liver stage
- b) Author contributions: Please remove from the manuscript and only specify author contributions in our submission system. CRediT has replaced the traditional author contributions section because it offers a systematic machine-readable author

contributions format that allows for more effective research assessment. You are encouraged to use the free text boxes beneath each contributing author's name to add specific details on the author's contribution. More information is available in our guide to authors:

<https://www.embopress.org/page/journal/17574684/authorguide#authorshipguidelines>

➤ Author contributions were removed and the CRediT system was updated

- c) Please correct the reference citations in the reference list - these should not be numerical but rather alphabetical. Where there are more than 10 authors on a paper, note that only 10 will be listed, followed by "et al.". DOIs should be removed for all publications. Please check "Author Guidelines" for more information.

<https://www.embopress.org/page/journal/17574684/authorguide#referencesformat>

➤ The reference list was modified according to the author guidelines.

- d) Data not shown: We do not allow statements/conclusions with "data not shown". As per our guidelines, on "Unpublished Data" the journal does not permit citation of "Data not shown". All data referred to in the paper should be displayed in the main or Expanded View figures. Please remove from page 24.

➤ This statement has now been removed.

2) In the Materials and Methods, please take care of the following:

- a) Cell lines: You have indicated in the Author Checklist that you used cell lines in the manuscript. Please include all information requested in the author checklist (accession number in repository or supplier name, catalog number, clone number, and/or RRID). Please also be sure to include a sentence in the Materials and Methods as to whether or not the cell lines were recently authenticated and tested for mycoplasma contamination - if this is not possible, please explain why in your point-by-point response to these requests.

➤ We actually did not use any "cell lines", so I have now selected the option "non applicable."

- b) The Materials & Methods section on *P. falciparum* parasite lines appears to be missing some details/incomplete (at the beginning there is the following "P. falciparum parasite lines and were cultured...")

➤ This section was modified to include more information, "Pf strain NF54 was used as the parent wildtype strain for all transgenesis and as the control strain in experiments. Pf NF54, Pf *mei2*⁻ and Pf parasite lines...."

c) Although you have indicated in the Author Checklist that "No blinding was done" has been included in the manuscript, this does not appear to be in the Experimental Animals section. Please ensure that a statement on whether or not blinding was done is included in the Materials & Methods even if no blinding was done.

- A statement for this has now been added to the Experimental animal section, "The assignment of which animals would be infected with wildtype vs transgenic parasites was performed randomly. Further downstream processing, including harvesting blood and tissue samples were not performed in a blinded fashion."

d) Antibodies: please ensure that company name, catalog number, and dilutions/amounts of each antibody are reported in the Materials & Methods. Currently only catalog and clone numbers are given for Flow Cytometry and some of the antibody information is missing from the Immunofluorescence assay section

- Antibody concentrations have been added for the antibodies used in flow cytometry.
- For anti-*Plasmodium* antibodies, catalog number was already included for the only commercial antibody used, Pf CSP clone 2A10. The other antibodies were either generated by the Kappe lab over the years or were obtained from the European Malaria Repository.
- The Plasmodium antibodies against Py ACP, *P. vivax* mHSP70, Py mTIP and Pf Exp1 were generated in the Kappe lab. The references for these antibodies have now been added to the materials and methods.
- Antibodies from the European malaria reagent repository, such as the Pf MSP1 12.1 antibody do not have a catalog number. The link to the Repository and the original paper in which it was first used is mentioned, as required by the database.

3) Please place individual sections of the manuscript in the following order: Title page - Abstract & Keywords - Introduction - Results - Discussion - Materials & Methods - Data Availability - Acknowledgements - Disclosure and Competing Interests Statement - The Paper Explained - For More Information - References - Figure Legends - Expanded View Figure Legends.

- The sections in the manuscript have been organized according to above.

4) Please move the Materials & Methods after the Discussion.

- The Materials & Methods section was moved after the Discussion.

5) Please incorporate the 'Study Approval' section into the Materials & Methods.

- The 'Study Approval' was incorporated into the Materials & Methods section.

6) The Paper Explained: Please add "The Paper Explained" to the main manuscript text.

- The Paper explained was added to the main text after the Disclosure and Competing Interests Statement

7) For more information: This space should be used to list relevant web links for further consultation by our readers. Could you identify some relevant ones and provide such information as well? Some examples are patient associations, relevant databases, OMIM/proteins/genes links, author's websites, etc...

- This section was added.

8) For the figures and figure legends, please take care of the following:

- a) Please upload the main figures and Expanded View figures as individual, high resolution figure files. Check the "Author Guidelines" for more information:
<https://www.embopress.org/page/journal/17574684/authorguide#figureformat>
 - High resolution figures and expanded view figures have been uploaded as per recommendations in the author guidelines.
- b) Please note that Panel F is not labelled in Figure 6
 - Panel F is now labeled in Figure 6.
- c) Appendix Figure S2 should be after Appendix Figure S1 in the Appendix file
 - Appendix Figure S1 and S2 are now in the correct order.
- d) Please note that a separate 'Data Information' section is required in the legends of figures 3c; 5a-e; 6b-c; EV 1b-e; EV 4b-c.
 - A separate data information section has been added to figures 3c; 5a-e; 6b-c; EV 1b-e; EV 4b-c.
- e) Please indicate the statistical test used for data analysis in the legend of figure 7c.

- Figure legend 7c has been updated with the statistical analysis used, which was unpaired t-test.
- f) Please note that in figures 1a; 6b-c; there is a mismatch between the annotated p values in the figure legend and the annotated p values in the figure file that should be corrected.
 - This mismatch in the p-values for figures 1a; 6b-c was corrected.
- g) Please note that information related to n is missing in the legends of figure 7c; EV 1b-e.
 - This information was added to legends of figure 7c; EV 1b-e.
- h) Please note that the error bars are not defined in the legends of figure 7c; EV 1b-e.
 - This information was added to legends of figure 7c; EV 1b-e.
- i) Please note that we require exact p-values to be reported. Currently exact p-values are not provided.
 - All figures now mention exact p-values.

9) Please ensure that all funding sources are entered into the manuscript submission system - it is unclear whether the National Institute of Allergy and Infectious Diseases (NIAID) be added to the list of funders.

- NIH-NIAID should be included as the funding source.

10) Synopsis image: the resolution of the synopsis image is not high. Is this sufficient for you? Otherwise please reupload in the same dimensions but with higher resolution.

- A new version of the synopsis file will be uploaded with a higher resolution.

11) Synopsis text: Please check your synopsis text before submission with your revised manuscript. Please be aware that in the proof stage minor corrections only are allowed (e.g., typos).

- Please use the passive voice (edit last sentence of standfirst).
- Please check the first bullet point - should both "exhibited" and "suffered" be included?).
- Please check the formulation of the second bullet point (grammar seems awkward)

- The synopsis text has been revised according to the above comments. It now uses the passive voice and it was edited for grammar.

12) Appendix file: Please upload the Appendix as a single PDF and add page numbers to the Table of Contents.

- The appendix file is now a single PDF and page numbers have been added to the table of contents.

13) As part of the EMBO Publications transparent editorial process initiative (see our policy here: https://www.embopress.org/transparent-process#Review_Process), EMBO Molecular Medicine will publish online a Peer Review File (PRF) to accompany accepted manuscripts. This file will be published in conjunction with your paper and will include the anonymous referee reports, your point-by-point response and all pertinent correspondence relating to the manuscript. Let us know whether you agree with the publication of the PRF and as here, if you want to remove or not any figures from it prior to publication. Please note that the Authors checklist will be published at the end of the PRF.

- We agree with the publication of the PRF and all the figures included therein.

14) Please provide a point-by-point letter INCLUDING my comments as well as the reviewer's reports and your detailed responses (as Word file).

- I have included a detailed point-by-point response letter to all the editor's comments, and I have also included all the reviewer's comments.

7th Mar 2024

Dear Prof. Kappe,

We are pleased to inform you that your manuscript is accepted for publication and is now being sent to our publisher to be included in the next available issue of EMBO Molecular Medicine. We are also delighted to inform you that a News and Views was commissioned to highlight your article and should also be published in the same issue.

Yours sincerely,

Poonam Bheda, PhD
Scientific Editor
EMBO Molecular Medicine
